# Scaling laws of bacterial and archaeal plasmids

Rohan Maddamsetti [1,2,3] ✉, Irida Shyti[1,2], Maggie L. Wilson[1,2], Hye-In Son[1,2], Yasa Baig[4], Zhengqing Zhou[1,2], Jia Lu [1,2] & Lingchong You [1,2,5] ✉

The capacity of a plasmid to express genes is constrained by its length and copy number. However, the interplay between these parameters and their constraints on plasmid evolution have remained elusive due to the absence of comprehensive quantitative analyses. Here, we present 'Pseudoalignment and Probabilistic Iterative Read Assignment' (pseuPIRA), a computational method that overcomes previous computational bottlenecks, enabling rapid and accurate determination of plasmid copy numbers at large scale. We apply pseuPIRA to all microbial genomes in the NCBI RefSeq database with linked short-read sequencing data (4644 bacterial and archaeal genomes including 12,006 plasmids). The analysis reveals three scaling laws of plasmids: first, an inverse power-law correlation between plasmid copy number and plasmid length; second, a positive linear correlation between protein-coding genes and plasmid length; and third, a positive correlation between metabolic genes per plasmid and plasmid length, particularly for large plasmids. These scaling laws imply fundamental constraints on plasmid evolution and functional organization, indicating that as plasmids increase in length, they converge toward chromosomal characteristics in copy number and functional content.

Plasmids are extrachromosomal DNA elements, found ubiquitously across bacteria and archaea, that mediate the flow of genes within and across microbial communities. Plasmids play a role in how microbial populations rapidly adapt to novel selection pressures by amplifying the copy number of beneficial genes and promoting their spread by horizontal gene transfer[1–3]. In the context of human health, plasmids shape human microbiome dynamics[4–6], in particular by serving as critical vectors for the dissemination of antibiotic resistance[7]. Plasmids are also foundational to biotechnology, as they can be engineered to control recombinant gene expression and the behavior of cells, populations, and microbial consortia[8,9]. Plasmid length and copy number have a substantial impact on plasmid gene expression. Here, we define the plasmid copy number (PCN) as the number of plasmid copies per the longest chromosome per cell. Understanding the interplay between plasmid length and PCN is essential, as these

parameters[10] affect the molecular biology, ecology, and evolution of microbes. Such understanding also has applications for engineering microbial populations and communities[11].

Intuitively, we would expect that plasmid lengths and copy numbers constrain each other. For instance, a cell may have a limited capacity to accommodate additional genetic material beyond the chromosome, which would impose a tradeoff between PCN and length. If so, then increasing PCN would necessarily constrain the functional capacity of a plasmid, in terms of the number and functional diversity of the genes it carries. Small-scale data supports this hypothesis: high-copy-number plasmids are often small[6], while larger conjugative plasmids often have 1–2 copies per cell[1]. A qualitative inverse correlation between plasmid size and copy number was described for 11 plasmids found in a Bacillus thuringiensis strain[12]. Importantly, an analysis of 2292 enterobacterial plasmids by Shaw

[1]Center for Quantitative Biodesign, Duke University, Durham, NC, USA. [2]Department of Biomedical Engineering, Duke University, Durham, NC, USA. [3]Department of Biochemistry and Microbiology, Rutgers University, New Brunswick, NJ, USA. [4]Department of Bioengineering, Stanford University, Stanford, CA, USA. [5]Department of Molecular Genetics and Microbiology, Duke University School of Medicine, Durham, NC, USA. ✉e-mail: rohan.maddamsetti@rutgers.edu; lingchong.you@duke.edu

et al.[13] also revealed a quantitative inverse correlation between plasmid size and copy number, even though the quantitative nature of this correlation and its potential significance were not examined in depth in that work.

Despite these earlier contributions, comprehensive data on the distribution of PCN across Bacteria and Archaea still do not exist[14]. Therefore, it is not known whether a quantitative relationship between plasmid size and copy number holds across Bacteria and Archaea. The rapidly increasing amount of sequence data on plasmid-bearing microbes creates an opportunity to address these questions. A major technical challenge is that direct PCN calculations at scale require pairwise sequence alignment between thousands of sequencing datasets and reference genomes, a process that is computationally prohibitive[15–21]. The computational costs associated with sequence alignment thus represent a key bottleneck that restricts PCN computations from approaching the scope or scale of all microbial genomes.

To overcome this bottleneck, we developed Pseudoalignment and Probabilistic Iterative Read Assignment (pseuPIRA), which uses pseudoalignment to rapidly and accurately estimate PCNs across large datasets. By applying pseuPIRA to all complete genomes containing plasmids in the NCBI RefSeq database[22] with linked short-read sequencing data in the sequencing read archive (SRA)[23], we report the largest dataset on plasmid lengths and copy numbers to date. Our analysis encompasses 4644 bacterial and archaeal genomes and 12,006 plasmids, spanning Bacteria and Archaea. We discovered universal scaling laws governing plasmid biology: first, an inverse power-law correlation between PCN and plasmid length; second, a positive linear correlation between protein-coding genes and plasmid length; and third, a positive correlation between metabolic genes per plasmid and plasmid length, particularly for large plasmids. These scaling laws imply fundamental constraints on the evolution of plasmids, as well as their ability to accommodate functional traits. Our findings reveal that as plasmids increase in length, they converge toward chromosomal characteristics in copy number and functional content, challenging traditional distinctions between plasmids and chromosomes. This discovery not only advances our understanding of plasmid dynamics and microbial evolution but also has implications for biotechnology, such as the rational design of synthetic plasmids and the engineering of microbial communities.

## Results

### PCNs are rarely reported in the microbial genomics literature
One goal of this project was to generate a comprehensive dataset of PCNs, because we found that few plasmids had reported copy numbers in the literature. To quantitatively assess this research gap, we randomly sampled 50 genomes, each containing at least one multicopy plasmid with PCN > 10 (Methods). We manually examined the publications associated with each genome, based on the genome annotation files found in the NCBI RefSeq database. PCNs were reported for 3 out of 50 genomes (Supplementary Table 1). Therefore, we estimate that ~6% of genomes with sequenced plasmids have PCNs that are reported in the literature.

### Pseudoalignment and Probabilistic Iterative Read Assignment (pseuPIRA) is a scalable and accurate method for PCN estimation
To address this critical gap and enable a comprehensive understanding of plasmid dynamics across Bacteria and Archaea, we developed Pseudoalignment and Probabilistic Iterative Read Assignment (pseuPIRA), which is described in Fig. 1A and Box 1. Using pseuPIRA, we estimated PCNs at an unprecedented scale, facilitating the discovery of universal scaling laws in plasmid biology. Our final dataset comprises 4644 bacterial and archaeal genomes with a total of 12,006 PCN estimates.

A direct PCN calculation only requires two assumptions. First, we assume that the lengths of all chromosomes and plasmids in the genome are known. Second, we assume that the relative amounts of sequencing data that map to chromosomes versus plasmids in each sequencing sample are proportional to the physical amount of DNA corresponding to chromosomes and plasmids in the genome. Figure 1A shows an example calculation. Suppose a genome has one chromosome and one plasmid with three copies relative to the chromosome. By dividing the total amount of sequencing data (in units of nucleotide base pairs) mapped to a chromosome or plasmid by the corresponding lengths of the chromosome or plasmid, the ratio of plasmid DNA to chromosomal DNA in the sequencing data can be calculated. This estimates the PCN per chromosome in the sequenced sample.

The direct PCN estimation method, however, does not account for sequencing reads that map to multiple replicons. Here, we define "replicon" as a generic term for either chromosomes or plasmids. We define a "uniread" as a sequencing read that unambiguously maps to a single replicon, and a "multiread" as a sequencing read that maps to multiple replicons. Multireads can arise due to repetitive or duplicated sequences that are shared across replicons. Such a situation can arise when plasmids and chromosomes share mobile genetic elements, such as a transposon that has jumped from the chromosome to a plasmid[2,3]. Multireads may affect PCN estimates when a plasmid shares significant homology with either the chromosome or other plasmids in the cell. In this case, an unknown fraction of multireads may come from the plasmid of interest, while the remainder comes from other replicons in the genome. However, that unknown fraction depends on the PCN, introducing a circular dependency.

pseuPIRA solves the multiread problem. Pseudoalignment is first used to map reads to replicons. Unireads are used to make an initial estimate of PCNs. The multireads are then re-aligned to the reference genome using traditional pairwise sequence alignment[24,25]. The multireads that map to a single genomic location with traditional alignment are combined with the unireads to improve the PCN estimates (Box 1). The remaining multireads are then probabilistically allocated to each replicon in the genome, based on the initial PCN estimates. The estimates are iteratively updated until convergence, based on the reallocation of multireads. That is, our pipeline uses pseudoalignment to quickly make good initial PCN estimates and then uses probabilistic iterative read assignment (PIRA) to refine those estimates based on multiread information.

Pseudoalignment overcomes the computational bottleneck that would be caused by using traditional alignment to map terabytes worth of short-read sequencing data to thousands of reference microbial genomes[15–21,26,27]. We compared the computational performance of our standalone Python implementation of pseuPIRA (https://github.com/rohanmaddamsetti/pseuPIRA) against CoverM, a state-of-the-art program for PCN estimation[28] implemented in Rust[29]. On a small genomic dataset (1.58 Gb data), pseuPIRA is 1.29× slower than CoverM (16.6 s versus 12.9 s), while on a large genomic dataset (90.6 Gb data), pseuPIRA is 1.67× faster (705.0 s versus 1175.7 s), demonstrating the superior computational scaling properties of pseudoalignment over alignment for PCN estimation (Supplementary Table 2).

We used pseuPIRA to estimate the copy number of 12,006 plasmids (Supplementary Data 1). Summary statistics for plasmids binned into percentiles by length are provided in Supplementary Data 2. This dataset represents the largest and most comprehensive set of PCNs to date, by comparison to the 6327 plasmids reported by Ramiro-Martinez et al.[28] and the 2292 plasmids reported by Shaw et al[13]. In addition our data span bacterial and archaeal taxa, while these previous datasets only cover plasmids from enterobacteria[13,28]. By comparing PCN estimates made with pseuPIRA to PCN estimates by the direct method (pseudoalignment only), we find that pseuPIRA

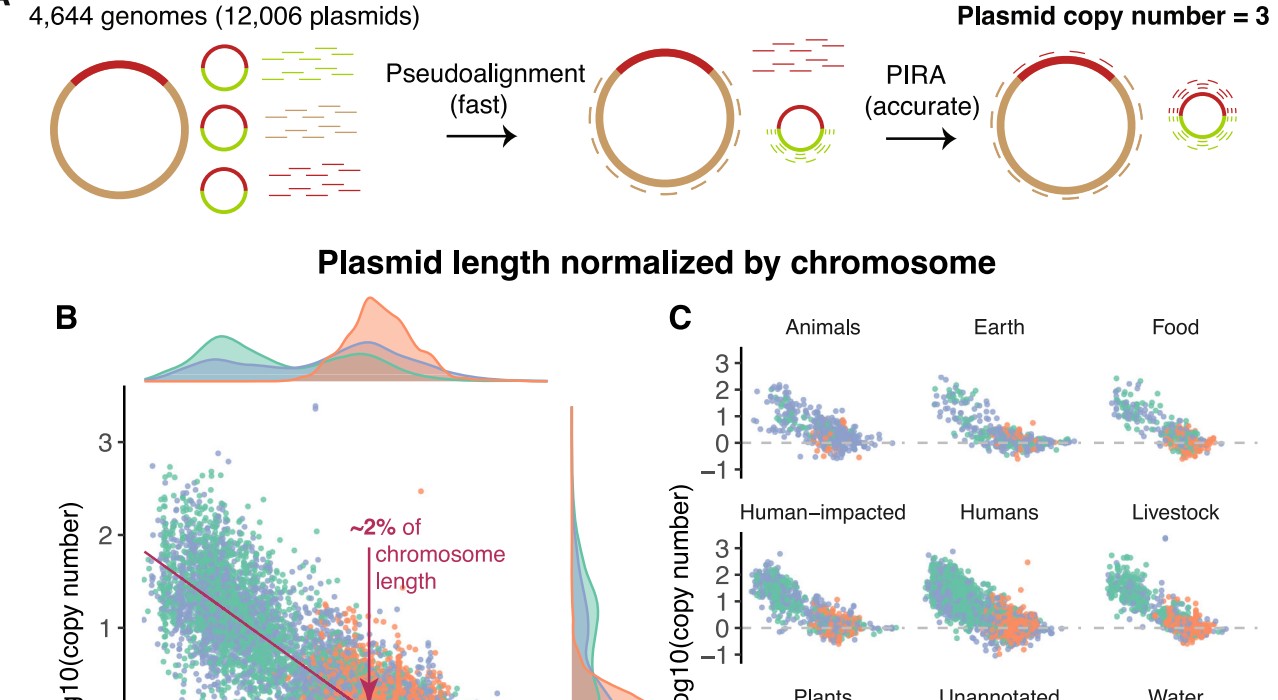

## Pseudoalignment and Probabilistic Iterative Read Assignment (pseuPIRA)

**Fig. 1 | Computation of plasmid copy numbers over bacteria and Archaea reveals that copy number inversely correlates with plasmid length. A** The computational pipeline. Suppose a genome has one chromosome and one plasmid with three copies relative to the chromosome. By dividing the total amount of sequencing data mapped to a chromosome or plasmid by the length of the chromosome and plasmid, the ratio of plasmid DNA to chromosomal DNA can be calculated. This estimates the plasmid copy number per chromosome in the sample. Pseudoalignment is used to rapidly estimate plasmid copy numbers, and Probabilistic Iterative Read Assignment (PIRA) is used to incorporate reads that map to multiple replicons (e.g., the chromosome and plasmid) to further improve plasmid copy number estimates. **B** Plasmid length inversely correlates with plasmid copy number. Rescaling plasmid length by the length of the largest chromosome in the cell reveals a scaling law. A segmented regression (in maroon) was fit to these normalized data on a log-log plot. The segmented regression has a first slope of −0.880, a breakpoint at −1.735 (or 1.84% of a chromosome), a second slope of −0.125, and an Adjusted $R^2$ of 0.690. The marginal density distributions of plasmid copy number and normalized length are displayed on the axes. **C** The inverse correlation holds across diverse environments. The ecological provenance of each replicon was annotated per the method described in Maddamsetti et al.[3] (Methods). Source data are provided as a Source Data file.

recovers PCN for 103 more plasmids, by enabling estimation for plasmids with many multireads but few unireads (Pearson correlation $\rho = 0.959$, Supplementary Fig. 1A). When unireads are abundant, then pseuPIRA generates PCN estimates that are consistent with the direct approach, which neglects multireads (Pearson correlation $\rho = 0.997$, Supplementary Fig. 1B). Several plasmids had estimated PCN < 1, meaning that the number of plasmid copies were lower than the number of chromosome copies in the sample sent for genome sequencing. To assess pseuPIRA's accuracy, we first compared PCN estimates generated by pseuPIRA to PCN estimates generated by traditional alignment algorithms[24,25,30]. Due to the computational overhead of using alignment to estimate PCNs, we selected a random set of 100 genomes to benchmark pseuPIRA against traditional alignment methods. Each of these randomly selected 100 genomes contained at least one plasmid with an estimated PCN < 0.8, to test whether these low PCN estimates were also recovered by traditional alignment algorithms. If so, this outcome would indicate that these low PCN

estimates were a property of the underlying sequencing data, and not an artifact caused by pseuPIRA.

We used two methods to estimate PCN using traditional alignment. First, we used minimap2, as a state-of-the-art method for pairwise alignment of sequencing reads to reference chromosomes and plasmids[24,25]. Second, we used breseq[30], an established genome resequencing pipeline that uses Bowtie 2[31] to align sequencing reads to reference chromosomes and plasmids. Importantly, both minimap2 and breseq have been used to estimate PCNs[2,3,32,33]. The comparison between pseuPIRA and minimap2 (Pearson correlation $\rho = 0.998$, Supplementary Fig. 1C) and between pseuPIRA and breseq (Pearson correlation $\rho = 0.991$, Supplementary Fig. 1D) shows that PCN estimates generated by pseuPIRA are consistent with PCN estimates generated by traditional read alignment algorithms. As an additional technical control, we examined whether PCN estimates generated by pseudoalignment were sensitive to the specific choice of software used. We compared PCN estimates generated by Themisto[27] (the

## BOX 1
# Probabilistic iterative read assignment

Definitions: a replicon is a general term for a chromosome or plasmid. A uniread is a sequencing read that maps to a unique replicon. A multiread is a sequencing read that maps to multiple replicons (e.g., the chromosome and one or more plasmids).

**PIRA initialization step:**

- Use pseudoalignment (*themisto* computer program) to assign sequencing reads to replicons.

- Make the initial PCN estimate vector $\boldsymbol{\pi_1}$ with the unireads: $\boldsymbol{\pi}_1 = \frac{1}{(R_1/L_1)} \begin{bmatrix} R_1/L_1 \\ \dots \\ R_j/L_j \\ \dots \\ R_n/L_n \end{bmatrix}$ where $R_j$ is the number of unireads mapping to replicon $\boldsymbol{j}$, and $\mathbf{L}$ is

  the length of replicon $\boldsymbol{j}$. We assume that the 1$^{st}$ replicon is the longest chromosome in the genome, with copy number 1.

- Map the multireads to the reference genome using alignment (*minimap2* computer program).

- Multireads that align to a unique genomic location are added to the set of unireads. $\boldsymbol{\pi_1}$ is updated to include this information.

- For each of the remaining (true) multireads:

  Get the number of matches to each replicon in the reference genome.
  Update the match matrix $\mathbf{M}$, where $\mathbf{M}_{ij}$ = the number of matches of multiread $\boldsymbol{i}$ to replicon $\boldsymbol{j}$.

  For example, suppose $\mathbf{M} = \begin{bmatrix} 1 & 2 & 0 & 1 \\ 0 & 1 & 1 & 0 \\ 1 & 1 & 0 & 0 \\ \dots & \dots & \dots & \dots \\ 0 & 2 & 1 & 0 \end{bmatrix}$. Each row of $\mathbf{M}$ represents a multiread.

Each column of $\mathbf{M}$ represents a replicon (there are 4 replicons in this genome). The first row represents a multiread that maps to replicon 1 once, replicon 2 twice, replicon 3 zero times, and replicon 4 once.

**PIRA iterations:**

- Turn the PCN estimate vector $\boldsymbol{\pi_1}$ into a diagonal matrix $\mathbf{D} = \mathrm{diag}(\boldsymbol{\pi_1})$.

- Weight each column of $\mathbf{M}$ by the corresponding entry of the PCN estimate vector $\boldsymbol{\pi_1}$. We do so by multiplying $\mathbf{M}$ by $\mathbf{D}$ to make the matrix-matrix product $\mathbf{MD}$.

- Normalize each row of the matrix $\mathbf{MD}$ to sum to one to make matrix $\mathbf{M}^*$ (This is the probabilistic read assignment step).

- Sum over rows of $\mathbf{M}^*$ to generate the multiread vector $\mathbf{R_D} = [R_{D1}\ R_{D2}\ \dots\ R_{Dn}]$.

- Use the multiread vector $\mathbf{R_D}$ to update $\boldsymbol{\pi_1}$ as follows: $\boldsymbol{\pi}_2 = \frac{1}{(R_1 + R_{D1})/L_1} \begin{bmatrix} (R_1 + R_{D1})/L_1 \\ \dots \\ (R_j + R_{Dj})/L_j \\ \dots \\ (R_n + R_{Dn})/L_n \end{bmatrix}$

- Iterate until the PCN estimate vector $\boldsymbol{\pi}$ converges–if this process takes $k$ iterations, then $\boldsymbol{\pi}_{k+1} \cong \boldsymbol{\pi}_k$ at convergence.

---

software used for pseuPIRA) against PCN estimates generated by kallisto[16]. As expected, we found that these PCN estimates were highly consistent (Pearson correlation $\rho = 0.991$, Supplementary Fig. 1E). Finally, we compared PCN estimates generated by pseuPIRA to the previously published PCN estimates in the Supplementary Table 2 of Shaw et al.[13] (Pearson correlation $\rho = 0.997$, Supplementary Fig. 1F). Together, these results indicate that pseuPIRA provides reliable and consistent PCN estimates, and show that the low PCN estimates are a property of the underlying sequencing data, given the consistency of these PCN estimates across methods and with previously published PCN data.

### A universal inverse power-law correlation between plasmid length and PCN

Using pseuPIRA, we generated the largest dataset on PCNs to date, covering genomes across Bacteria and Archaea. This dataset reveals an inverse power-law correlation between PCN and plasmid length (Fig. 1B, Supplementary Figs. 2 and 3, Supplementary Data 2). The distribution of plasmid sizes is bimodal, so K-means clustering with K = 2 was used to assign plasmids into two clusters. The cluster of small plasmids has a mean copy number of 28.1 plasmids per chromosome with a standard deviation of 68.4, and a range from 0.08 to 2433 copies

per chromosome. The cluster of small plasmids has a mean length of 6579 bp with a standard deviation of 4811.9 and a range from 1025 to 22,729 bp. The cluster of large plasmids has a mean copy number of 1.89 plasmids per chromosome with a standard deviation of 5.3 and a range from 0.10 to 305.2 copies per chromosome. The cluster of large plasmids has a mean length of 141,607 bp with a standard deviation of 179,864.6 bp, and a range from 22,780 to 2,586,495 bp (Supplementary Data 3).

MOB-typer[34] was used to classify plasmids as conjugative, mobilizable, or non-mobilizable. 3181 out of 3203 conjugative plasmids fall into the cluster of large plasmids (99.3%). By contrast, mobilizable plasmids (i.e., those that can be transferred by conjugation, but do not themselves encode conjugation machinery) and non-mobilizable plasmids have a wide distribution of lengths and may fall into either the large or small clusters (Fig. 1B). 229 plasmids are longer than 500,000 bp in length. Among these 229 megaplasmids[35], 31 are conjugative, 28 are mobilizable, 103 are non-mobilizable, and 67 are unannotated, consistent with previous observations that most megaplasmids are non-mobilizable[36]. Many of these megaplasmids are chromids, which resemble secondary chromosomes even though they replicate using plasmid replication and partitioning systems[35,37].

This inverse correlation between PCN and plasmid length is universal, holding across diverse environments (Fig. 1C and Supplementary Fig. 3) and bacterial and archaeal taxa (Supplementary Fig. 4). Furthermore, this inverse correlation between PCN and length largely holds within individual genomes as well. Out of 2810 genomes containing two or more plasmids, 2301 have an inverse correlation between plasmid length and copy number (mean Pearson correlation coefficient = −0.92), while 509 show a positive correlation (mean Pearson correlation coefficient = 0.86). Out of 1724 genomes containing three or more plasmids, 1539 have an inverse correlation between plasmid length and copy number (mean Pearson correlation coefficient = −0.87), while 185 show a positive correlation (mean Pearson correlation coefficient = 0.61). The intragenomic inverse correlation between PCN and length is even stronger when we consider the 2142 genomes that contain at least one small plasmid. 1764 of these genomes contain two or more plasmids, and 1689 of those have an inverse correlation between plasmid length and copy number (mean Pearson correlation coefficient = −0.93), while 75 show a positive correlation (mean Pearson correlation coefficient = 0.75). 1323 of these genomes contain three or more plasmids, and 1287 of those have an inverse correlation between plasmid length and copy number (mean Pearson correlation coefficient = −0.90), while 36 show a positive correlation (mean Pearson correlation coefficient = 0.47). This analysis suggests that the genomes with positive intragenomic correlations between plasmid length and copy number are largely genomes that only contain large plasmids. It is possible that many such positive correlations could occur by chance, assuming that these plasmids largely have PCN ~ 1.

We normalized the length of each plasmid by the length of the longest chromosome in its genome (Fig. 1B, C). These data are well fit by a segmented regression model[38,39] in which PCN linearly scales with plasmid length on a log-log scale, up to a length threshold at which PCNs start to converge to chromosomal copy numbers (Fig. 1B, Supplementary Figs. 2 and 3). The breakpoint for this scaling law occurs when the plasmid reaches 1.8% of the length of the chromosome. A model comparison using Akaike's Information Criterion shows that the segmented regression model (AIC = 8152.2, $R^2$ = 0.690) is significantly better than both a linear regression model (AIC = 9614.6, $R^2$ = 0.649) and a second-order polynomial regression model (AIC = 8400.6, $R^2$ = 0.683). The same pattern holds when a segmented regression model is fit to the unnormalized data: the segmented regression (AIC = 8048.8, $R^2$ = 0.692) is significantly better than both a linear regression model (AIC = 9824.2, $R^2$ = 0.643) and a second-order polynomial regression model (AIC = 8212.6, $R^2$ = 0.688). The breakpoint for the segmented regression fit to the unnormalized length data occurs at 56,624 bp. Further details about the segmented regression are provided in Supplementary Table 3.

The segmented regression model suggests the following interpretation. PCNs can be modeled as a mixture of plasmids with cell-cycle-dependent and cell-cycle-independent replication mechanisms. The low-copy number conjugative F plasmid (length 99,159 bp) has 1–2 copies per cells, and replicates in sync with the cell cycle in *Escherichia coli*[40–42]. By contrast, the multicopy R6K plasmid (39,872 bp) replicates in a cell-cycle independent manner[41,43,44]. The scaling law may represent how PCN scales with plasmid length for plasmids that replicate using cell-cycle independent mechanisms. Once plasmids reach a critical length threshold, which seems to be ~2% of the length of the chromosome, mechanisms that coordinate plasmid replication with cell division become critical for stable plasmid maintenance.

### Small multi-copy plasmids mostly coexist with large, low-copy plasmids

Plasmids often co-occur with other plasmids in the environment[45], and positive interactions between plasmids can stabilize such co-existence within cells[46]. We asked how often multi-copy plasmids co-existed with larger plasmids in these data. Out of 4644 genomes containing plasmids with PCN estimates, 1834 contain a single plasmid. By contrast, out of 1440 genomes containing plasmids with PCN > 10 in our data, only 184 contained those multi-copy plasmids as their sole plasmid. Therefore, plasmids with PCN > 10 mostly coexist with large low-copy plasmids (Binomial test: $p < 10^{-15}$). Interactions among plasmids, in particular, interactions among small multi-copy plasmids and larger conjugative plasmids, may therefore play a role in the empirical scaling law between plasmid length and copy number.

### Genetic features associated with plasmid length and copy number

**The inverse power-law between plasmid length and PCN holds across plasmid taxonomic units (PTUs).** We assigned plasmids to PTUs using multiple, previously published plasmid classifications, to see how plasmid lengths and copy number vary with genetic distance. First, we used the PTUs reported by Acman et al.[47] and Redondo-Salvo et al.[48], who clustered plasmids into PTUs by $k$-mer similarity and Average Nucleotide Identity, respectively. In both cases, plasmids within a given PTU cluster by length, and have similar copy numbers (Supplementary Fig. 5). Second, we clustered plasmids into PTUs based on similarity by Mash distance (using the default 0.06 threshold) using MOB-cluster[34]. Third, we typed plasmids based on their Rep proteins with MOB-typer[34]. Again, plasmids within PTUs based on these definitions have similar lengths and copy numbers. Finally, we used the Rep typing reported by Ares-Arroyo et al.[49]. Since plasmid incompatibility groups are largely defined by the Rep proteins that initiate plasmid replication, these results indicate that PCN strongly associates with plasmid length and the molecular systems that initiate plasmid replication. Across these plasmid classification systems, plasmid length is more conserved than PCN within PTUs. This finding indicates that the copy numbers for small plasmids can vary over an order of magnitude, while highly related plasmids (as defined by K-mer similarity, and ANI) have similar lengths (Supplementary Fig. 5). It is likely that the conservation of plasmid length within PTUs, to some degree, is related to the fact that PTUs are often defined using metrics that are highly sensitive to plasmid length (such as $k$-mer similarity, Mash distance, and ANI). Regardless, our analysis shows that the universal inverse power-law correlation between plasmid length and PCN holds across PTUs as defined by several different methods.

**Plasmid relaxase typing does not determine plasmid length and PCN.** Plasmid lengths and copy numbers are not determined by plasmid mobility types as annotated by MOB-typer[34], although some mobility groups are specific to large conjugative plasmids. Plasmid mobility groups are defined by the relaxase proteins that nick plasmids at *oriT* transfer origins to initiate horizontal gene transfer by conjugation. Specifically, many mobility groups show two modes, one corresponding to small mobilizable multi-copy plasmids, and a second corresponding to large low-copy conjugative plasmids (Supplementary Fig. 6).

**No correlations between PCN and plasmid host range.** We also examined plasmid lengths and copy numbers in the context of host range annotation made by MOB-typer[34] and the host range plasmid annotations reported by Redondo-Salvo et al.[48] (Supplementary Fig. 7). Large and small plasmids are found together across annotated host ranges, indicating that plasmid size and copy number does not correlate well with plasmid host range.

### Many plasmids have PCN < 1

Our data shows that many plasmids have a lower copy number than the chromosome (Fig. 1, Supplementary Fig. 8). Our methodological validations of pseuPIRA (Supplementary Fig. 1) show that this result is a genuine property of the sequencing data used for these PCN estimates.

Out of the 12,006 plasmids in these data, 2487 plasmids have PCN < 1, representing 21% of all plasmids. Across ecological categories, between 10% and 25% of plasmids have PCN < 1 (Supplementary Fig. 8B).

## High copy number plasmids are rare and are enriched in human-impacted environments

High copy number plasmids (PCN > 50), by contrast, are relatively rare in this dataset (Supplementary Fig. 7A, C). Out of the 12,006 plasmids in these data, 537 plasmids have PCN > 50, representing 4.5% of all plasmids. High copy number plasmids are significantly enriched in human-impacted environments (Supplementary Fig. 8B). Together, these observations suggest that small multicopy plasmids tend to have stable copy numbers of ~10–40 copies per chromosome per cell (Fig. 1), such that plasmids with very high copy numbers (PCN > 50) may be signatures of recent positive selection for higher plasmid gene expression[50].

## Functional properties and organization of plasmids approach the functional organization of chromosomes as they increase in size

Our findings show how PCNs converge to chromosome copy numbers, as plasmids increase in length (Fig. 1). We hypothesized that many functional properties of plasmids should converge to the functional properties of chromosomes as they increase in size. Indeed, we find that scaling laws emerge for the fraction of protein-coding sequences per plasmid and for the number of metabolic proteins per plasmid, as plasmids increase in length. These scaling laws converge to scaling laws that hold for chromosomes. This analysis was conducted on all complete microbial genomes containing plasmids in the NCBI RefSeq database, comprising 18,253 genomes containing 48,569 plasmids at the time of analysis.

**Protein-coding sequence scaling law.** Smillie et al.[36] reported that larger plasmids have protein-coding densities approaching that of chromosomes, while small plasmids are less coding dense. Our data shows that this pattern is universal across environments and microbial taxa. As plasmids increase in size, the fraction of sequence dedicated to protein-coding sequences converges to the fraction of sequence dedicated to protein-coding sequences on chromosomes (Fig. 2A, B). This pattern holds across plasmids and chromosomes sampled across diverse environments and bacterial and archaeal taxa (Fig. 2C and Supplementary Fig. 9). This finding implies that as plasmids decrease in length, protein coding density also decreases, as previously indicated by Smillie et al.[36]. A simple explanation is that the overhead for regulatory sequences that determine plasmid replication, stability, and maintenance is relatively larger for small plasmids compared to large plasmids. For instance, a plasmid replication origin may take up some fixed length of DNA, and the relative length of this noncoding sequence decreases as a plasmid increases in length. In other words, as a plasmid increases in length, the relative length of the minimal sequence requirements for an autonomously replicating plasmid decreases, while this fraction may be relatively large for very small plasmids (Fig. 2B).

**Emergence of a metabolic scaling law as plasmids approach chromosome length scales.** We annotated metabolic genes on plasmids by mapping genes to the metabolic pathways annotated in the KEGG database[51], using the GhostKOALA functional annotation webserver[52]. Plasmid size scales with the number of metabolic genes on the plasmid. Given the computational cost of annotating chromosome metabolic genes with the GhostKOALA webserver, we only annotated metabolic genes for 100 chromosomes, arbitrarily chosen to span the full rank distribution of chromosomes by length. The metabolic scaling relationship found for these 100 chromosomes emerges among megaplasmids that are longer than 500,000 bp

(Fig. 3). Again, this emergent scaling law holds across plasmids and chromosomes sampled across diverse environments and bacterial and archaeal taxa (Fig. 3B and Supplementary Fig. 10). This scaling law may emerge due to fundamental constraints on cellular energetics. Mega-plasmids may need more metabolic capacity (measured by the number of metabolic genes) to compensate for the metabolic burden required to maintain such large plasmids. Strikingly, the metabolic scaling law only fails for Mycoplasmatota (Supplementary Fig. 10B); these bacteria are obligate pathogens that lack cell walls and include bacterial species with the smallest known cells, genomes, and metabolisms, even lacking key pathways like the TCA cycle[53]. This finding additionally supports our interpretation that the metabolic scaling law is caused by fundamental constraints on cellular energetics.

## Discussion

We have uncovered universal scaling laws that govern fundamental aspects of plasmid biology, revealing deep-rooted principles underlying microbial evolution and genome organization. This work reports the largest dataset on PCN and length to date, encompassing 4644 bacterial and archaeal genomes and 12,006 plasmids. Our discovery of an inverse power-law correlation between PCN and plasmid length indicates a fundamental constraint that dictates how much plasmid DNA can be stably transmitted over time: once plasmid DNA content reaches ~2% per chromosome, plasmid replication needs to be synchronized with the cell cycle for stable inheritance. This scaling law not only advances our understanding of plasmid biology but also challenges traditional distinctions between plasmids and chromosomes, suggesting a continuum of genetic elements shaped by fundamental biophysical constraints.

One surprising finding in our analysis is the prevalence of plasmids with PCN < 1. Why may this be the case? First, the PCN estimates reported in this work are relative to chromosome copy numbers, and do not represent estimates of the absolute number of plasmids per cell. It is known that bacteria can contain multiple chromosome copies per cell[54–56], such that a clonal bacterial population can contain subpopulations with different numbers of chromosome copies per cell. The frequencies of these subpopulations can change over long-term evolution[57]. Furthermore, some large bacteria show extreme polyploidy comprising tens of thousands of chromosome copies per cell[58]. This means that PCN estimates <1 can arise in cases where, say, a bacterium has four chromosome copies and two plasmid copies. In this scenario, we would estimate PCN = 0.5. Second, PCN estimates <1 could arise when genomic DNA is prepared from exponential-phase cultures. During exponential growth, PCNs can be lower than chromosomal copy numbers[59,60]. Third, this result could be caused by plasmid heterogeneity in sequenced bacterial clones, where some, but not all, daughter cells contain the plasmid. In other words, the mixture of cells in an otherwise clonal sample may vary in the number of plasmids per cell, where the plasmid is present in some cells and absent from others. Such a scenario could arise during dynamic division of labor on plasmids, where a costly plasmid is maintained in a bacterial population by horizontal gene transfer from slow-growing producer cells to fast-growing cells without the plasmid[61]. PCN < 1 could also be maintained by the presence of parasitic satellite plasmids that effectively reduce the copy number of the plasmid of interest[62].

While we were finalizing our manuscript, we became aware of independent work reporting both a large dataset of PCNs as well as a scaling law relating PCN to plasmid length[28]. The convergence of findings from independent studies validates the universality of the inverse scaling law between plasmid length and copy number. That said, the variance in PCN for small plasmids spans two orders of magnitude for the same normalized length, and this variance in PCN decreases as plasmid length increases (Fig. 1 and Supplementary Data 2). Understanding why PCNs show such variability around the inverse scaling law remains an open question in need of further

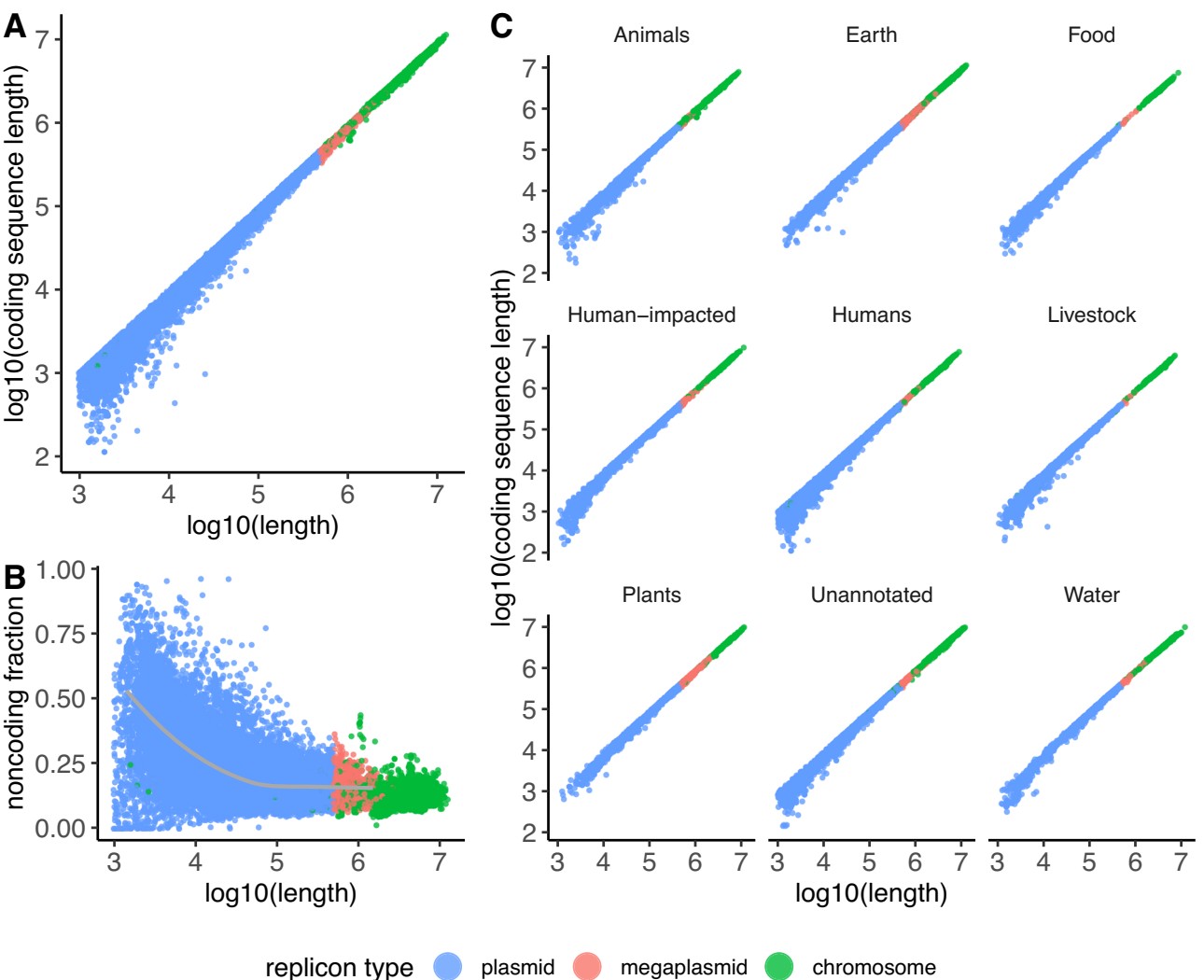

**replicon type** ● plasmid ● megaplasmid ● chromosome

**Fig. 2 | Protein-coding sequences on plasmids follow an empirical scaling law.** Comparison of scaling between chromosomes and plasmids on a log-log scale. Plasmids are shown in blue, megaplasmids (plasmids > 500,000 bp in length) are shown in red, and chromosomes are shown in green. Source data are provided as Source Data files. **A** As plasmids increase in size, the fraction of sequence dedicated to protein-coding sequences converges to the fraction of sequence dedicated to protein-coding sequences on chromosomes. **B** As plasmids increase in size, the fraction of sequence dedicated to protein-coding sequences increases. The noncoding fraction is defined as (length − coding sequence length) / (length). The gray line indicates the average noncoding fractions and lengths of 100 buckets of plasmids, binned by length. **C** The same pattern holds for microbes sampled across diverse environments. The ecological provenance of each replicon was annotated per the method described in Maddamsetti et al.[3] (Methods).

investigation. Significant variability in PCN estimates may be caused by differences in bacterial growth conditions, DNA extraction methods[14,63], and experimental protocol (e.g., qPCR or whole genome sequencing[64]), so for this reason, we note that any particular PCN estimate in our dataset should be interpreted with care. Regardless, we expect our finding of an inverse scaling law between PCN and plasmid length to be robust, given that it holds across 12,006 PCN estimates sampled across diverse bacteria and environments, and collected by diverse research groups using a range of experimental protocols for bacterial growth and DNA sequencing. We also expect the inverse scaling law to be robust to phylogenetic non-independence between genomes and plasmids, because the inverse scaling law holds across PTUs (Supplementary Fig. 4). That said, an ongoing technical challenge in the field is the development of methods that can explicitly take phylogenetic non-independence between plasmids into account, given the plastic and dynamic nature of plasmid evolution which is dominated by structural variation, recombination, and horizontal gene transfer over point mutations and vertical inheritance[65,66].

Our analyses also reveal scaling laws constraining fundamental aspects of plasmid biology, such as protein-coding regions and metabolism. We found that as plasmids increase in size, their functional properties and organization converge toward those of chromosomes, highlighting a unifying principle in genome evolution. These findings are consistent with previous reports that the functional gene content of microbial genomes follow empirical scaling laws[67–72], and show that such relations extend to plasmids. The emergent metabolic scaling law, in particular, indicates that larger plasmids incorporate more metabolic genes, suggesting that metabolic capacity is a critical factor in the stable maintenance of very large plasmids and chromids. Together, our results demonstrate the existence of biophysical or evolutionary constraints that dictate stable plasmid coding structures. Such constraints may include bioenergetic or metabolic factors, or fundamental trade-offs between plasmid replication and transcription[73,74], that (1) constrain the copy number of large plasmids, (2) select for lower coding density for small plasmids, and (3) select for increased numbers of metabolic genes, once plasmids pass some critical length threshold.

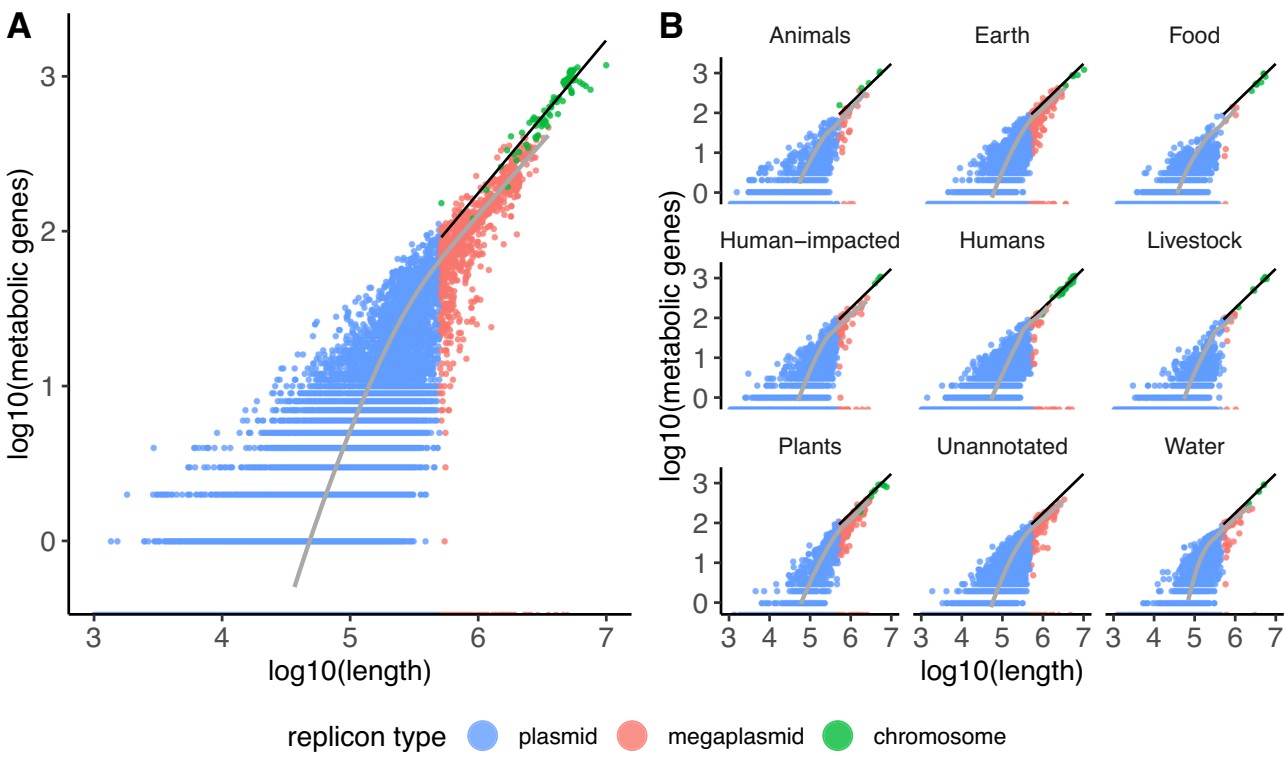

replicon type ● plasmid ● megaplasmid ● chromosome

**Fig. 3 | A metabolic scaling law emerges as plasmids approach chromosome length scales.** Comparison of scaling between chromosomes and plasmids on a log-log plot. Plasmids are shown in blue, megaplasmids (plasmids > 500,000 bp in length) are shown in red, and chromosomes are shown in green. The linear regression between $\log_{10}$(metabolic genes) and $\log_{10}$(length) for chromosomes is shown in black. The gray line indicates the average length of plasmids containing $n$ metabolic genes, where $n$ is an arbitrary integer. As $n$ increases, the scaling between plasmid length and the number of metabolic genes $n$ (shown in gray) approaches the scaling for chromosome length and number of metabolic genes $n$ (shown in black). Source data are provided as a Source Data file. **A** Plasmids can vary in size by orders of magnitude but still carry similar numbers of metabolic genes—but as plasmids reach megaplasmid scales (>500,000 bp in length), their metabolic gene content begins to scale like chromosomes. **B** The emergent metabolic scaling law holds for microbes sampled across diverse environments. The ecological provenance of each replicon was annotated per the method described in Maddamsetti et al.[3] (Methods).

These constraints have broad implications for the evolution of microbial genomes, the dynamics of horizontal gene transfer, and the adaptability of microbial communities in various environments.

The PCN-length scaling law could suggest an approximate conservation of the fraction of cellular resources allocated for the replication and maintenance of multicopy extrachromosomal DNAs. This scaling law could also emerge from multileveled selection on plasmid replication[75], by at least two mechanisms. First, Takeuchi et al. report a scaling law of multilevel evolution, in which the size of a collective (i.e., intracellular PCN) $N$ and the mutation rate of its components $m$ satisfy the relation $Nm^{\alpha} = K$, where the scaling exponent $\alpha$ depends on multiple factors and $K$ is constant[76]. If the mutation rate scales linearly with plasmid length, then this model predicts an inverse scaling law between PCN and length. Second, Xue et al.[77] combine multilevel selection with a coarse-grained model of microbial metabolism to predict that optimal plasmid size inversely scales with PCN. For the simplest cases, these mechanisms all predict an inverse scaling between plasmid length and copy number with a slope of −1 on a log-log scale. However, our data point to a more nuanced picture (Supplementary Table 3 and Supplementary Fig. 11). Suppose the inverse scaling relation between plasmid length and copy number has slope = −1 on a log-log scale. Then plasmid DNA content (PCN × plasmid length) should be constant (i.e., a flat line with slope = 0 on a log-log scale). However, Supplementary Fig. 11 shows that the segmented regression on normalized plasmid DNA content has a first slope of 0.118 and a second slope of 0.747 (Adjusted $R^2 = 0.316$). Therefore, further theory and quantitative experiments are needed to understand the biophysical and evolutionary origins of these empirical scaling laws.

Our findings have practical implications for biotechnology and synthetic biology. Understanding the scaling laws governing plasmid biology can inform the rational design of synthetic plasmids, optimizing them for desired functions while ensuring stable inheritance and minimal metabolic burden on host cells[11,33]. One intriguing implication of the scaling laws we uncovered is that larger plasmids may serve as a more efficient chassis for engineered functions. As plasmid size increases, the fraction of DNA dedicated to protein-coding sequences and metabolic genes scales predictably, converging toward chromosomal characteristics. This suggests that larger plasmids have greater potential to accommodate diverse functional modules, particularly those related to metabolic processes. By contrast, smaller plasmids appear to require a higher fraction of their noncoding DNA for essential functions such as replication, stability, and maintenance (Fig. 2B). This observation could indicate that small plasmids are more constrained in their ability to host engineered gene circuits, as the available space for protein-coding sequences is more limited. This scaling behavior raises the possibility that larger plasmids are inherently more efficient for applications where high encoding capacity is required. Not only can they support more protein-coding sequences, but they may also be more suited to carry complex metabolic pathways. By using larger plasmids as chassis, synthetic biologists might achieve more stable and scalable designs with reduced competition between essential functions and engineered traits. This finding could help guide the selection of plasmid sizes in the design of synthetic constructs, favoring larger plasmids for applications requiring extensive metabolic or functional gene integration[78,79].

Moreover, our work also demonstrates the power of applying efficient algorithms (e.g., pseudoalignment) developed for one problem (transcriptomics), to another (PCN estimation) to reveal new biology. For example, pseudoalignment could accelerate the inference of microbial growth rates from peak-to-trough coverage ratios in microbiome data[80], as well as the discovery of structural variation in microbiomes[81], as the state-of-the-art currently depends on alignment-based methods[80,81]. Based on our current results, we hypothesize that such an analysis would show that plasmid replication rates converge to chromosome replication rates in microbiome data, as plasmid lengths approach the size of chromosomes. Second, pseudoalignment should accelerate the discovery of high-copy-number plasmids and gene amplifications in large databases of metagenome-assembled genomes (MAGs)[82–84]. By rapidly remapping raw sequencing data to MAGs using PIRA or other pseudoalignment-based methods, the relative copy numbers of different replicons in a metagenomic sample may be estimated. This approach could facilitate the identification of genetic elements associated with antibiotic resistance, virulence factors, or metabolic capabilities, with significant implications for public health and environmental microbiology. The discovery of replicons with elevated copy number in microbiome data may represent high-copy-number plasmids, mobile genetic elements, viruses, or genetic amplifications mediated by selection and horizontal gene transfer[2,3]. Finally, pseudoalignment and PIRA have potential applications to biological questions beyond PCN estimation, as gene copy number changes are important in both eukaryotic genome evolution (including plant domestication) and cancer evolution.

## Methods

### PCN analysis

**Input genomes for the PCN analysis pipeline.** A table of microbial genomes in the NCBI genomes database was downloaded from: https://ftp.ncbi.nlm.nih.gov/genomes/GENOME_REPORTS/prokaryotes.txt (last accessed March 17, 2025). This table was filtered for genomes containing plasmids by running the following UNIX command: grep "plasmid" data/prokaryotes.txt | grep "Complete Genome" | sed 's/GCA/GCF/g' > results/complete-prokaryotes-with-plasmids.txt. This command restricts the analysis to high quality complete genome assemblies that contain at least one annotated plasmid. This resulted in a filtered table of 19,211 bacterial and archaeal genomes containing plasmids. These genomes from the NCBI RefSeq database were then filtered by the availability of corresponding Illumina short-read sequence data in the NCBI SRA.

**Downloading of reference genome annotation and corresponding Illumina short-read sequencing data.** Data downloading was automated with the Python 3.12 script PCN_pipeline.py. First, metadata for each genome in prokaryotes-with-chromosomes-and-plasmids.txt was examined to find the subset of genomes with paired-end Illumina sequencing data in the NCBI SRA. This resulted in a table, RunID_table.csv, containing records for 4853 genomes containing plasmids. The NCBI RefSeq database was cross-checked against these genomes, and reference genome annotation data were downloaded for the subset of microbial genomes found in the NCBI RefSeq database. Illumina paired-end short-read sequencing data in fastq format was downloaded for each of these microbial genomes in RefSeq; this step was the most time-consuming step of this pipeline, taking two weeks to download ~15 TB of sequencing data, using the prefetch and fasterq-dump programs in NCBI SRA Toolkit v3.2.0[23]. At this stage, a Python 3.12 script called check-genome-quality-and-consistency.py was used to ensure that all downloaded fastq data corresponded to a downloaded reference genome, and to ensure that all reference genomes contained complete chromosome assemblies. In all, sequencing data for 4849 microbial genomes in NCBI RefSeq were downloaded.

**Pseudoalignment of sequencing reads against reference genomes and direct PCN estimation.** Sequencing data processing was automated in the Python 3.12 script PCN_pipeline.py. kallisto 0.51.0 and themisto 3.2.2 were used to pseudoalign Illumina sequencing reads against reference chromosomes and plasmids. The sequencing reads were successfully pseudoaligned to chromosomal and plasmid reference sequences in 4834 genomes. Plasmids with fewer than 10,000 mapped reads were removed from the analysis, resulting in a final set of 4644 genomes containing 12,006 plasmids. We do not take sequencing read quality into account; one benefit of pseudoalignment is that with very high probability, reads with errors and sequencing artifacts such as adapters or barcodes will fail to pseudoalign to any replicon, and are therefore omitted from the analysis[16,27]. Sequencing coverage per replicon (i.e., chromosome or plasmid) was estimated by dividing the number of reads mapping to the replicon by the length of the replicon. Then, direct estimates of PCNs (relative to the chromosome) were generated by dividing the mean sequencing coverage for each plasmid by the mean sequencing coverage for the largest chromosome in each genome. Sequencing read length is omitted from this calculation, as this is a constant factor that cancels out when dividing plasmid coverage by chromosome coverage.

**Probabilistic iterative read assignment.** PIRA was developed to further increase the accuracy of PCN estimates by incorporating information from sequencing reads that map ambiguously to multiple replicons within a genome, such as a read that maps to both a plasmid and chromosome. This situation can arise when reads come from repeated or duplicated sequences, as in the case of a read that corresponds to a transposon found in multiple locations in a genome. A specification of the PIRA algorithm is provided in 1, and PIRA is implemented in the Python 3.12 code pseuPIRA.py (https://github.com/rohanmaddamsetti/pseuPIRA). For a given genome, an initial copy number estimate vector is generated using the direct method described in the previous section. This initial copy number estimate vector ignores multireads that pseudoaligned to multiple replicons. The original fastq files are then filtered for multireads, and these multireads are re-aligned to the reference genome using minimap2 version 2.29-r1283[25]. Reads that align to a single location by traditional sequencing alignment with minimap2 are tabulated. These are used to further improve the initial copy number estimate vector. The remaining $m$ multireads are then tabulated into an $m \times n$ matrix, where $n$ is the number of replicons in the genome, sorted by length, such that the longest replicon (that is, the main chromosome) is in the first position. Each row of the matrix (each row corresponding to a single multiread) has entries corresponding to the number of times that this multiread maps to each replicon in the genome.

The columns of this matrix are scaled by the entries of the initial copy number vector, so that each row (corresponding to a single multiread) accounts for both the number of locations on the replicon that align with this multiread, as well as the current copy number estimate of each replicon. Each row is then normalized to sum to one, such that each row now represents the probability distribution of which replicon that multiread came from. Then, all rows are summed to make a $1 \times n$ vector that sums to $m$; this vector represents how all the multireads probabilistically map to the replicons. This vector of multiread counts is used to update the copy number estimate vector (by adding these multiread counts to the number of unireads mapping to the replicon, and dividing the total by replicon length), and the process is iterated until the copy number estimate vector converges (i.e., the norm of the change between iterations falls below a small value like $10^{-6}$).

**Mathematical underpinnings of PIRA.** PIRA is a special case of the Expectation-Maximization (EM) algorithm[85,86]. The EM algorithm has previously been coupled with pseudoalignment to estimate transcript

counts from RNA-seq data[16,19] and species abundance in metagenomic data[17,26]. Suppose we have $m$ sequencing reads and $n$ replicons. Each sequencing read stems from a unique, but possibly unknown, replicon DNA sequence. In a complete dataset, each sequencing read $r_j$ would be associated with a unique label $\eta_j$ indicating what replicon it came from. Given multinomial sampling, the likelihood of the PCN vector $\boldsymbol{\pi}$ (where $\boldsymbol{\pi_1} = 1$ = chromosome copy number, and $\boldsymbol{\pi_2}, ..., \boldsymbol{\pi_n}$ are PCNs) given complete data (a vector of $m$ sequencing reads $\boldsymbol{r}$ and an associated vector of true replicon labels $\boldsymbol{\eta}$) has the form: $\mathscr{L}(\boldsymbol{\pi}|\boldsymbol{r},\boldsymbol{\eta}) = \frac{m!}{r_1!...r_k!...r_m!} \prod_{j=1}^{m} P(r_j, \eta_j|\boldsymbol{\pi})$. Here, $P(r_j, \eta_j|\boldsymbol{\pi})$ is the probability of sampling read $j$ from replicon $k$ given our current PCN estimate $\boldsymbol{\pi}$. $P(r_j, \eta_j|\boldsymbol{\pi}) = \frac{\pi_k \cdot length_k}{\sum_{k=1}^{n} \pi_k \cdot length_k}$, which is the fraction of DNA originating from replicon $k$ given its length $length_k$ and its copy number $\pi_k$. The problem is that multireads map to multiple replicons, so the specific replicon that a multiread came from is ambiguous. Our goal, therefore, is maximum likelihood estimation of PCNs from incomplete data[85,87]. We can solve this problem using the EM algorithm for multinomial count data, as described in the classic Dempster-Laird-Rubin paper[85]. We can think of the complete data as a $m \times n$ matrix $\mathbf{L}$ whose $(j, k)$ element = 1 if sequencing read $j$ comes from replicon $k$, and zero otherwise. The $j$th row contains $n-1$ zeros and one entry = 1. However, if sequencing read $j$ is a multiread, then we can estimate the complete data for the row for sequencing read $j$ by letting the value of cell $\mathbf{L}_{j,k}$ be the probability that sequencing read $j$ came from replicon $k$. This is the E-step of this EM algorithm, in which we estimate the complete data matrix $\mathbf{L}$ given our current estimate of the PCN vector $\boldsymbol{\pi}$. Note that the assigned probabilities in $\mathbf{L}_{j,k}$ are also expected values, since the expectation of an indicator function is simply its probability. We then make the M-step of the EM algorithm, in which we make the maximum likelihood estimate for the PCN vector $\boldsymbol{\pi}$, given our current estimate for the complete data matrix $\mathbf{L}$. We then iterate the E-step and M-step until $\boldsymbol{\pi}$ converges, as guaranteed by the theory in Wu[86]. It is possible for multireads to map to multiple locations *within* a replicon due to sequence repeats or duplications. To take this into account, we define the match matrix $\mathbf{M}$, where $\mathbf{M}_{j,k}$ is the number of distinct locations that sequencing read $j$ aligns to replicon $k$. So, we define $\mathbf{L}_{j,k} = \frac{\mathbf{M}_{j,k} \cdot \pi_k}{\sum_{k=1}^{n} \mathbf{M}_{j,k} \cdot \pi_k}$ to take the multiplicity of sequence matches to each replicon into account. Note that this only matters for multireads, since for a uniread $j$, $\mathbf{L}_{j,k} = 1$, no matter the value of $\mathbf{M}_{j,k}$. Therefore, we only need to use sequence alignment to calculate $\mathbf{M}_{j,k}$ for multireads. These considerations lead to the following equation to update the vector of PCN estimates $\boldsymbol{\pi}$ from PIRA iteration $i$ to $i+1$: $\pi_k^{(i+1)} = \frac{\sum_{j=1}^{j=m} \mathbf{L}_{j,k}}{length_k} \cdot \frac{1}{\frac{\sum_{j=1}^{j=m} \mathbf{L}_{j,1}}{length_1}}$

where the second term normalizes the PCN vector $\boldsymbol{\pi}$ so that $\boldsymbol{\pi_1} = 1$ = chromosome copy number. We can speed up this calculation by treating unireads and multireads separately, so that $\pi_k^{(i+1)} = \frac{R_k + \sum_{j=1}^{j=m^*} \mathbf{L}_{j,k}}{length_k} \cdot \frac{1}{\frac{\sum_{j=1}^{j=m} \mathbf{L}_{j,1}}{length_1}}$, where $R_k$ is the number of unireads that pseudoaligned to replicon $k$, $m^* = m - R_k$ is the number of multireads in the data, and $R_{Dk} = \sum_{j=1}^{j=m^*} \mathbf{L}_{j,k}$ is the number of multireads that were probabilistically assigned to replicon $k$ in the $i^{th}$ PIRA iteration.

**Comparison with the ICRA algorithm.** At a high level, PIRA is in the same class of methods as the iterative coverage-based read assignment (ICRA) algorithm described by Zeevi et al.[81], but differing in four aspects. First, PIRA was designed to infer copy numbers within genomes, and not within metagenomes. Second, PIRA lets a multiread contribute to multiple replicons based on the probability that the read originated from that replicon, rather than assigning a multiread to the single best match (soft read assignment, rather than hard read assignment, in machine learning parlance). Third, PIRA weights the

probability distribution of how a multiread maps to replicons within a genome by (1) the number of matches of that read to a given replicon and (2) the current estimate of that replicon's copy number; unlike ICRA, alignment read quality is not considered. Fourth, ICRA divides replicons into genomic bins, and iteratively estimates the copy numbers of each genomic bin to find copy number variation; by contrast, PIRA treats each replicon as a single bin. In principle, PIRA could be generalized to estimate copy number within genomic bins like ICRA: this is a direction for future research. While PIRA was conceived independently from ICRA, its similarities to ICRA gave us confidence that PIRA was the correct approach for incorporating multiread information. In addition, the two-dimensional matrix data structure used to store multireads for PIRA was conceived as a simpler version of the 3-dimensional array described by Zeevi et al.[81] in their formal description of ICRA.

**Benchmarking of PCN estimates.** We benchmarked our PCN estimate pipeline using a random subset of 100 genomes, each containing at least one plasmid with an estimated PCN < 0.8. This benchmarking was automated in the Python 3.12 script PCN_pipeline.py. First, PCNs were re-estimated on these 100 genomes using minimap2 version 2.29-r1283 to align all reads, rather than using pseudoalignment. PIRA was used to account for multireads found with minimap2. Second, PCNs were re-estimated on these 100 genomes using breseq 0.37 to align all reads. In this case, multireads were ignored, and PCNs were estimated by dividing plasmid mean sequencing coverage by chromosome mean sequencing coverage. We also compared PCN estimates with the direct method (ignoring multireads) using themisto 3.2.2 to pseudoalign reads, and PCN estimates using PIRA on top of the direct method using themisto. We also compared direct PCN estimates using Themisto 3.2.2 with direct PCN estimates using kallisto 0.51.0 to see how the choice of pseudoalignment software affected PCN estimates, if at all. Finally, we benchmarked our PIRA PCN estimates against the PCN estimates published in Supplementary Table 2 of Shaw et al.[13]. The python 3.12 script parse_REHAB_plasmids.py was used to preprocess metadata for this comparison.

**Ecological annotation**
The ecological provenance of each microbial genome was annotated per the method described in Maddamsetti et al.[3]. Briefly, we used the host and isolation_source fields in the RefSeq annotation for each genome to place each into the following categories: Marine, Freshwater, Human-impacted (environments), Livestock (domesticated animals), Agriculture (domesticated plants), Food, Humans, Plants, Animals (non-domesticated animals, also including invertebrates, fungi and single-cell eukaryotes), Soil, Sediment (including mud), Terrestrial (non-soil, non-sediment, including environments with extreme salinity, aridity, acidity, or alkalinity), and Unannotated (no annotation). For reproducibility, our annotations are generated using a Python 3.12 script called annotate-ecological-category.py and checked for internal consistency using a Python 3.12 script called check-ecological-annotation.py. To simplify the data presentation, we merged categories as follows. Marine and Freshwater categories were grouped as Water. Sediment, Soil, and Terrestrial categories were grouped as Earth. Plant and Agriculture categories were grouped as Plants.

**Estimation of PCN reporting in the scientific literature**
A random seed was drawn between 1 and 100 using the random number generator at www.random.org[88]. For replicability, this random seed (the number 60) was used to draw 50 genomes without replacement, from a set of 1216 genomes containing at least one plasmid with an estimated PCN greater than 10. The RefSeq annotation files for each of these genomes were examined manually for publications associated with the given genome assembly. Each publication was

examined manually to see if PCNs were reported. The results of this analysis are reported in Supplementary Table 1.

### Plasmid scaling law analysis

**Input genomes for the plasmid scaling law analysis.** A table of microbial genomes in the NCBI genomes database was downloaded from: https://ftp.ncbi.nlm.nih.gov/genomes/GENOME_REPORTS/prokaryotes.txt. This table was filtered for complete microbial genomes containing plasmids by running the Python 3.12 script filter-genome-reports.py. Reference genome annotation files for these genomes were downloaded using the Python 3.12 script fetch-gbk-annotation.py. Then, the following Python 3.12 scripts were run to tabulate ecological metadata, protein counts and lengths for each genome: make-chromosome-plasmid-table.py, make-gbk-annotation-table.py, and count-proteins-and-replicon-lengths.py.

**Protein-coding sequence calculations.** The fraction of protein-coding sequences per replicon was calculated in two ways. First, by summing up the length of each protein-coding sequence in each replicon (i.e., plasmid or chromosome) in each genome, and dividing by the length of that replicon. Second, by counting the number of unique sites in protein-coding sequences in each replicon (to avoid double-counting sites in overlapping genes) and dividing by the length of the replicon. These calculations were run using the Python 3.12 script calculate-CDS-rRNA-fractions.py. 180 plasmids with no protein-coding sequences were removed from the analysis, as manual checks against the NCBI RefSeq database showed that some of these cases were clear annotation artifacts.

**Analysis of metabolic genes on plasmids.** We annotated metabolic genes on plasmids following the computational protocol in Hamrick et al.[61]. Briefly, the GhostKOALA functional genomics web server[52] (https://www.kegg.jp/ghostkoala/) associated with the Kyoto Encyclopedia of Genes and Genomes (KEGG) database[51] was used to annotate plasmid proteins with KEGG Orthology (KO) IDs. The Python 3.12 scripts make-plasmid-protein-FASTA-db.py and make-plasmid-GhostKOALA-input-files.py were used to generate input files for GhostKOALA. Each of these input files was manually submitted to the GhostKOALA web server at https://www.kegg.jp/ghostkoala and saved to disk. Then, the shell script concatenate-and-filter-plasmid-Ghost-KOALA-results.sh was used to concatenate the corresponding Ghost-KOALA output files and filter for plasmid proteins that were successfully mapped to a KEGG KO ID. The Python 3.12 script remove-chromosomes-from-plasmid-Ghost-KOALA-results.py was used to remove any proteins found on plasmids larger than chromosomes, as these were assumed to be annotation errors. The union of all KEGG KO IDs found among the plasmid genes was generated using the Python 3.12 script get_unique_KEGG_IDs.py. The output of this script was uploaded to the KEGG Mapper Reconstruct web server at: https://www.kegg.jp/kegg/mapper/reconstruct.html. The set of plasmid KEGG KO IDs that mapped to metabolic pathways (KEGG PATHWAY Database ID: 01100) was saved to file. Then, the Python 3.12 script get-plasmid-metabolic-KOs.py was run to generate a table of all metabolic genes on plasmids.

**Analysis of metabolic genes on chromosomes.** For comparison to the analysis of metabolic genes on plasmids, we examined metabolic genes on the chromosomes of 100 representative and arbitrarily chosen genomes. Genomes were ranked based on the length of their chromosome, and 100 genomes were chosen, by virtue of being roughly equally distributed across the rank distribution of chromosome lengths, over the set of genomes with ecological annotation (i.e., genomes marked as "Unannotated" were never chosen), and the proteins found in these 100 genomes were put into a small database file

using the Python 3.12 script make-chromosome-protein-FASTA-db.py. This input file was submitted to the GhostKOALA web server, and the GhostKOALA output file was saved to disk. Then, the shell script filter-chromosome-GhostKOALA-results.sh was used to filter the Ghost-KOALA output for chromosomal proteins that were successfully mapped to a KEGG KO ID. The Python 3.12 script get_unique_chromosome_KEGG_IDs.py was used to get the union of all KEGG IDs found among these chromosomal proteins. These data were uploaded to the KEGG Mapper Reconstruct webserver (https://www.kegg.jp/kegg/mapper/reconstruct.html), and the set of chromosomal KEGG KO IDs that mapped to metabolic pathways (KEGG PATHWAY Database ID: 01100) was saved to file. Then, the Python 3.12 script get-chromosome-metabolic-KOs.py was used to generate a final table of all KEGG metabolic proteins in the sampled chromosomes.

### Plasmid typing metadata

All plasmids were annotated using MOB-typer 3.1.7[34] using the protocol reported by Hamrick et al.[61]. Specifically, a Python 3.12 script called write-plasmid-seqs-for-MOB-typer.py was used to generate input files for MOB-typer. MOB-typer was automated with a Python 3.12 script called run-MOB-typer.py, and this Python script was run on the Duke Compute Cluster using a sbatch shell script called run-MOB-typer.sh as follows: sbatch run-MOB-typer.sh. MOB-typer results for each plasmid were combined into a single text file called combined_mob_typer_results.txt, using a Python 3.12 script called comb_mob_genome_data.py. Duplicate MOB-typer results were identified by running the following one-line shell script: sort combined_mob_typer_results.txt | sort | uniq -d > duplicated.txt. Then, the number of duplicate entries was counted with this one-line shell script: cat duplicated.txt | wc -l. Then, a final text file without duplicate entries was generated with the following one-line-shell script: sort -u combined_mob_typer_results.txt -o unique_mob_results.txt. Finally, plasmid mobility predictions from MOB-typer were parsed with a Python 3.12 script called parse-MOBtyper-results.py. The plasmid mobility annotations used in this work were previously reported in Supplementary Data File 5 of Maddamsetti et al.[3]. Plasmids were also typed using the supplementary data reported by Acman et al.[47], Redondo-Salvo et al.[48], Coluzzi et al.[89], and Ares-Arroyo et al.[49].

### Statistical analysis

All statistical analysis and data visualizations were generated using an R 4.2 script called PCN-analysis.R. At this stage, plasmid sequences <1000 bp in length were removed from the analysis to remove unmapped or unplaced plasmid contigs. Genomes with plasmids longer than their chromosome were also removed to remove potential genome misannotation errors.

### Reporting summary

Further information on research design is available in the Nature Portfolio Reporting Summary linked to this article.

## Data availability

All data analyzed in this project were retrieved from the NCBI RefSeq and SRA databases and are therefore publicly available. All processed data in this study are provided in the Source Data provided with this paper. Source data are provided with this paper.

## Code availability

A Github repository containing a standalone implementation of pseuPIRA, including a test example, is available at: https://github.com/rohanmaddamsetti/pseuPIRA (https://doi.org/10.5281/zenodo.15668298). A Github repository containing all data and code sufficient to reproduce the PCN estimation pipeline, including downloading of genomic data, is available at: https://github.com/rohanmaddamsetti/PCN-db-pipeline (https://doi.org/10.5281/zeno

do.15668288). A GitHub repository containing all computer codes for the remaining analyses, including statistics and figures, is available at: https://github.com/rohanmaddamsetti/plasmid-scaling-laws (https://doi.org/10.5281/zenodo.15668292).

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

## Acknowledgements

We thank the members of the You lab for helpful discussions and comments, and Duke Research Computing for technical assistance and computing resources. This work is partially supported by the National Institutes of Health (R01AI125604, R01GM098642, R01EB031869, L.Y.). The funders had no role in study design, data collection and analysis, decision to publish, or preparation of the manuscript.

## Author contributions

R.M. and L.Y. conceived and designed the research. R.M., I.S., M.L.W., H.S., and Z.Z. conducted bioinformatic analyses. J.L. checked PIRA for

correctness. R.M., I.S., Z.Z., and J.L. tested pseuPIRA and the data analysis pipeline. R.M., Y.B., and L.Y. wrote the manuscript. R.M., J.L., Z.Z., H.S., Y.B., and L.Y. edited the manuscript. L.Y. supervised research.

## Competing interests

The authors declare no competing interests.
