## [Transparent Peer Review file · Nature Communications]

Scaling laws of bacterial and archaeal plasmids

Corresponding Author: Professor Rohan Maddamsetti

Version 0:

Reviewer comments:

Reviewer #1

(Remarks to the Author)

The manuscript by Maddamsetti and colleagues describes a novel algorithm to compute the relative frequency of replicons by accounting more precisely for the coverage of reads that are multi-replicon. This is an interesting, albeit somewhat incremental, improvement on existing methods. The authors use this method to show that plasmid copy number (PCN) is inversely correlated with plasmid size. They call this a universal scaling law. They add two other scaling laws: a positive linear correlation between protein-coding genes and plasmid length and a positive correlation between metabolic genes per plasmid and plasmid length, particularly for large plasmids. The paper is not very well-written and contextualisation on existing literature requires a lot of improvement.

Major comments:

- The scaling law that gets more attention here, between PCN and plasmid size, has been known for a long time. For decades it has remained qualitative. But it was also quantified recently. For example, Shaw, Science Advances, 2021 shows the same exact scaling law in Figure 3A. This scaling law is not new and the relevant past papers are not cited.
- The fact that the number of genes correlates well with plasmid size is also well-known from works with over 20 years. Even the detail that for small size plasmids the gene density is lower was reviewed in reference #31 more than a decade ago (page 437).
- The text of the manuscript has several problems. Some examples here. The results section has many details that should be in Methods. The discussion needs improvement as it does not really discuss the biological reasons justifying the scaling laws and lacks references. Many paragraphs are too small, down to less than two lines. Section in page 10 pertains to discussion not results. Section "A scalable bioinformatics pipeline for PCN estimation" is not really about Results. Novel results only really start at page 3 of the Results section. The summary paragraph at the end of discussion seems redundant.
- The authors propose a method to identify PCN and compare it with other methods based on the same principle (sequence coverage). The difference between methods does not seem very large (Figure S1). There are some potential advantages of this method for lower PCN. But if the authors could provide coefficients of correlation for the panels in figure S1, they would reveal very high correlation. A related point is that there is no information on whether this method or the others are right. PCN was identified experimentally for some plasmids and it would be pertinent to know how they compare with published experimental data. By the end of page 17 the authors mention of a comparison with literature, but this table is never cited in results. When I went to see the table I found that for most of these plasmids there is no information for PCN, which makes it not very useful. Also, for those plasmids for which there are numbers on PCN, the numbers are not given (nor the comparison with the computed PCN).
- In page 8 there is a description of the fit of the model. But several important points seem missing. 1) if one has a plasmid of a given size, what is the expected interval of confidence for its copy number? This is a very relevant question that could be answered by this study. 2) How much of the variance is it explained by the scaling law? 3) What is the formula of the curve? This can be used to at least compute the expected PCN (although knowing the interval of confidence would be better).

Minor comments:

- "By contrast, mobilizable plasmids (i.e., those that can be transferred by conjugation, but do not themselves encode conjugation machinery) and non-mobilizable plasmids may be either large or small (Figure 1B)." This is well known.
- Page 10 last paragraph. I could not understand why this explanation justifies why high copy number plasmids were more frequent in human-impacted environments.
- Page 11. "This finding implies that as plasmids increase in length, protein coding becomes more efficient." I don't understand this statement. What does it mean exactly that protein coding becomes more efficient?
- Discussion. "Together, our results demonstrate the existence of fundamental constraints that dictate stable plasmid coding structures" Actually, the variance is quite high. The analysis of figure 1 reveals that PCN varies by two orders of magnitude

for the same normalised length. I would suggest to tone down the mention to "fundamental constraints".

- In methods. Please provide the date of accession to the database.

- First paragraph in page 15 is hard to follow. I suggest a more detailed text and the inclusion of mathematical formulae.

- In page 18. "The fraction of protein-coding sequences per replicon was calculated by summing up the length of each protein in each replicon (i.e., plasmid or chromosome) in each genome, and dividing by the length of that replicon" The length of each gene (not protein), I assume.

- It would be nice if the program had a downloadable example to be run that did not imply downloading 15 Tb of data (in present conditions, I couldn't do it to make a test).

(Remarks on code availability)

The presentation/documentation of the programs could be improved. The authors should also provide a toy example to test the program.

Reviewer #2

(Remarks to the Author)

Scaling laws of plasmids across the microbial tree of life

Maddamsetti et al. have developed a sensible framework for plasmid copy number evaluation and applied it to a wide range of publicly available microbial sequences. Overall I very much enjoyed reading this paper and the complementary companion paper by Ramiro-Martínez et al.

My one major query reading Maddamsetti et al. is that the text implies that their PIRA pipeline is a bioinformatic tool that others can apply to their own data, when in fact it appears to be a hard-coded pipeline to reproduce the results of the manuscript (something I think few will attempt, given the large resource requirements the authors point out in their methods). This work would be much more impactful if the PIRA pipeline was available for others to apply to their own data. The github repo (<https://github.com/rohanmaddamsetti/PCN-db-pipeline>) should also have more descriptive documentation of the pipeline itself - a graphic of the workflow and tools used would be useful.

Minor comments:

Abstract (and intro):

"The capacity of a plasmid to express genes is constrained by two parameters: length and copy number." - The phrasing of this sentence implies that gene expression is only due to length and copy number, however gene expression is also controlled by a number of other factors. I would rephrase to "plasmid length and copy number have a substantial impact on plasmid gene expression"

page 3, paragraph 1: "as they can engineered" -> typo, "as they can be engineered"

Page 7, paragraph 3: Please provide range and SD for small and large plasmid cluster copy numbers and lengths.

Page 7, paragraph 4: "Furthermore, this inverse correlation between PCN and length largely holds within individual genomes as well." For the genomes that had positive correlations between length and copy number - was there any insight into why this was? This was between 10-20% of the sample size, so not infrequent. Was there anything interesting about the microbial species/source or plasmid type?

page 9: "length is more conserved than copy number within PTUs"

Kmer, ANI and mash distances are very dependent on length, so I find this conclusion somewhat circular. While this statement may be true based on this analysis, I am not sure it is reflective of any meaningful biological interpretation.

The last sentence, "This finding indicates that highly related plasmids have similar lengths, but that the copy numbers for small plasmids can vary over an order of magnitude"

Again, 'highly related' is based on the classification system being used, which here is mostly methods that rely on conserved sequence similarity and would favour similar length plasmids (and is again a bit of a circular argument). The second half of the sentence doesn't address any conclusion to do with PTUs.

page 15: was there any QC of the sequencing reads for quality? Noting that PIRA does not take read quality into account

Page 15: the github repo (<https://github.com/rohanmaddamsetti/PCN-db-pipeline>) only lists Kallisto as a dependency, but Thermisto was used according to the manuscript? Please clarify. Please also list the tool dependencies and their version in the github.

Page 18: Protein-coding sequence calculations: did this account for overlapping genes?

Page 18: typo: "The, the union of all KEGG KO IDs"

Sup table 1: URL should be changed to DOI or pubmed ID, URL may become inactive over time

figure S6 (and other similar figures): add key on figure for green/orange/blue dots

(Remarks on code availability)

As above in main comments section. The code is available for reproducibility purposes, but not made to be easily accessible. The formatting of the readme in <https://github.com/rohanmaddamsetti/plasmid-scaling-laws> could be improved to help users follow the instructions. The instructions often refer to the authors host HPC, which is fine but should be distinguished from more general instructions to the wider community.

The PIRA pipeline (<https://github.com/rohanmaddamsetti/PCN-db-pipeline>) is not constructed as a stand-alone tool but as a hard-coded pipeline that will limit its accessibility to the community.

Reviewer #3

(Remarks to the Author)

The authors need to take into account the following major points:

#) sequencing data

The authors stated, "Applying PIRA to all microbial genomes in the NCBI RefSeq database with linked short-read sequencing data in the Sequencing Read Archive (SRA), we analyzed 4,317 bacterial and archaeal genomes encompassing 11,338 plasmids, spanning the microbial tree of life."

However, if the authors are unsure about the methods used to obtain these sequencing data (e.g., bacterial growth conditions and DNA extraction methods used), the reliability of the results could be compromised. This issue should be addressed or, at the very least, discussed.

Below are relevant excerpts from a reference paper:

<https://www.sciencedirect.com/science/article/pii/S2001037018301685>

Plasmid size may be associated with copy number. For 11 plasmids found in *Bacillus thuringiensis* strain YBT-1520, the plasmid sizes (ranging from 2 to 416 kb) and the copy numbers determined by quantitative polymerase chain reaction (ranging from 1.38 to 172) were negatively correlated [94].

Plasmid copy number estimates can vary, depending on bacterial growth conditions and DNA extraction methods used [97,99]. Therefore, copy number data should be interpreted carefully.

#) PIRA accurately estimates PCN

I suggest that the authors validate the accuracy of the PIRA-derived plasmid copy number (PCN) estimates by comparing them to the PCN values reported in previous studies, such as those cited below and in Supplementary Table 1:

Supplementary Table 1. Three out of fifty randomly chosen genomes containing plasmids have publications with reported plasmid copy numbers (PCN).

<https://academic.oup.com/bioinformatics/article/32/22/3380/2525610>

plasmidSPAdes: assembling plasmids from whole genome sequencing data

The average copy numbers (estimated as coverage ratios) varied from 1.4 for *Bce* to 14.0 for *Cfr*.

#) A universal inverse power-law correlation between plasmid length and PCN

In statistical analyses with multiple genomes (e.g., correlation between plasmid length and PCN), data points should not be assumed to be independent, as closely related lineages share many traits from their common ancestor.

Reference:

Joseph Felsenstein (1985) Phylogenies and the Comparative Method <https://www.jstor.org/stable/2461605>

Here are some minor points:

#) tree of life

The authors repeatedly use the term 'tree of life' throughout the manuscript. However, while taxonomic groups are presented in the paper, no phylogenetic trees—the 'tree of life' itself—are shown. If a phylogenetic tree is not provided or not considered, the term 'tree of life' should not be used.

(Remarks on code availability)

Version 1:

Reviewer comments:

Reviewer #1

(Remarks to the Author)

The authors did substantial work to improve the manuscript and tackle my comments. Thanks for that. This is an interesting contribution that is now clearer, better contextualized and the discussion is much richer. A few comments:

The level of quantification displayed along the results section is very variable and at places it is vague. One should not have to dig in the substantial supplementary material to have an idea of the quantities involved. Some examples (lines are approximative, since the PDF does not number all lines):

- Around line 165. Comparable performance is vague, and slightly misleading. "On a small genomic dataset (1.58 Gb data), pseuPIRA has comparable performance to CoverM. " And "On a large genomic dataset (90.6 Gb data), however, pseuPIRA is substantially faster". This should not have qualifications that are arguable, it should have numbers. On a small dataset pseudoPIRA is 28% slower, on a large dataset (where speed is more relevant) it is twice as fast.
- Line 175 "generates PCN estimates that are consistent". Please quantify with a correlation coefficient of something of the sort
- Line 195. " estimates generated by pseuPIRA are consistent with PCN estimates generated by traditional read alignment algorithms" Same comment
- Line 199 "estimates were highly consistent" same comment
- Line 201 "Finally, we compared PCN estimates generated by pseuPIRA to the previously published PCN estimates in the Supplementary Table 2 of Shaw et al " The result of the comparison is not mentioned in the main text.

Other comments:

- Line 307. I was confused with this new text. The authors state explicitly that PTUs conservation of size is likely a technical artifact, but show no evidence whatsoever of that. Previous works using more accurate methods (not ANI) have clearly demonstrated that plasmids within PTU have much more similar sizes than between PTUs. Which makes sense since plasmids with similarity are expected to have more similar sizes even if variance may be high. The authors own work goes in the same direction. If the authors want to draw some caution on the discussion about clustering of plasmids, that's fine. But here it's a clear statement in the Results section that lacks any kind of evidence. This may create great confusion in for newcomers in the field. As this is peripheral to the work, I suggest removing this unsubstantiated claim here and add a line that clustering of plasmids is hard and should be subject to caution in discussion.
- Discussion is still missing indications that comparisons here are being done between programs and not with actual experimental data. And that this and other approaches are assuming that coverage is an accurate measure of PCN.
- There is now more emphasis on plasmids with PCN <1. But there is a lot of variance here. I think this is missing a serious statistical analysis. What is the % of plasmids with PCN significantly lower than one? This number (not just those that are <1) should be the basis of discussion (which is very interesting).
- Sup Fig 1 is missing in the PDF I got.
- Line 475. The authors data does show a more nuanced view. Maybe this should be explained in a couple of lines, in order to make it explicit?

(Remarks on code availability)

I couldn't install the program simply because pip install uv didn't work in my computer and all the rest depends on it (MacOS X).

Reviewer #2

(Remarks to the Author)

(Remarks on code availability)

the authors have responded to previous comments and created a standalone tool for PCN estimates. While I have not installed/tested the tool, the github is well documented.

Reviewer #3

(Remarks to the Author)

(Remarks on code availability)

I recommend that the authors ensure the code runs correctly across different computing environments (Linux, macOS, etc.) and update the README documentation accordingly.

Below are the commands I executed and their outputs when testing the installation and setup instructions:

```
...
git clone https://github.com/rohanmaddamsetti/pseuPIRA
cd pseuPIRA/

# https://github.com/rohanmaddamsetti/pseuPIRA?tab=readme-ov-file#requirements

$uname -m
arm64

wget https://github.com/algbio/themisto/releases/download/v3.2.2/themisto-v3.2.2-aarch64-apple-darwin22.tar.gz
tar xvzf themisto-v3.2.2-aarch64-apple-darwin22.tar.gz
cp themisto-v3.2.2-aarch64-apple-darwin22/themisto ~/bin

# https://github.com/rohanmaddamsetti/pseuPIRA?tab=readme-ov-file#dependency-management-with-uv

$pip install uv
Collecting uv
Downloading uv-0.7.2-py3-none-macosx_11_0_arm64.whl.metadata (11 kB)
Downloading uv-0.7.2-py3-none-macosx_11_0_arm64.whl (15.5 MB)
----- 15.5/15.5 MB 18.3 MB/s eta 0:00:00
Installing collected packages: uv
Successfully installed uv-0.7.2

$uv install
error: unrecognized subcommand 'install'

tip: a similar subcommand exists: 'uv pip install'

Usage: uv [OPTIONS] <COMMAND>

For more information, try '--help'.
...

git clone https://github.com/rohanmaddamsetti/PCN-db-pipeline

# https://github.com/rohanmaddamsetti/PCN-db-pipeline?tab=readme-ov-file#setup

mkdir my-test-PCN-project
cd my-test-PCN-project
mkdir -p {data,results,src}

curl https://ftp.ncbi.nlm.nih.gov/genomes/GENOME_REPORTS/prokaryotes.txt > data/prokaryotes.txt ## on mac and linux

grep "plasmid" data/prokaryotes.txt | grep "Complete Genome" | sed 's/GCA/GCF/g' > results/complete-prokaryotes-with-plasmids.txt

conda create --name PCNdb_env --clone base
conda activate PCNdb_env
pip install pysradb biopython HTSeq beautifulsoup4 polars
conda install -c bioconda kallisto breseq
#Proceed ([y]/n)?
conda install -c conda-forge ncbi-datasets-cli --yes

# Install SRA-Toolkit:

# https://github.com/rohanmaddamsetti/PCN-db-pipeline?tab=readme-ov-file#running-the-pipeline

conda activate PCNdb_env
cd src/
sbatch --mem=16G -t 430:00:00 -p youlab --wrap="python PCN_pipeline.py"

-bash: sbatch: command not found
...
```

Responses to reviewers' comments

Reviewer #1 (Remarks to the Author):

The manuscript by Maddamsetti and colleagues describes a novel algorithm to compute the relative frequency of replicons by accounting more precisely for the coverage of reads that are multi-replicon. This is an interesting, albeit somewhat incremental, improvement on existing methods. The authors use this method to show that plasmid copy number (PCN) is inversely correlated with plasmid size. They call this a universal scaling law. They add two other scaling laws: a positive linear correlation between protein-coding genes and plasmid length and a positive correlation between metabolic genes per plasmid and plasmid length, particularly for large plasmids. The paper is not very well-written and contextualisation on existing literature requires a lot of improvement.

We appreciate the constructive feedback from Reviewer 1. We particularly appreciate their specific comments, which we have used to improve our writing and improve how our results are contextualized with existing literature, including Shaw et al. (2021).

We have added a benchmark (Supplementary Table 2) that compares the performance of our method for PCN estimation (pseudoalignment + probabilistic iterative read assignment) to the CoverM software used by Ramiro-Martinez et al. 2025. Our method has comparable performance on small sequencing datasets (1.58 Gb data) and is substantially faster on large sequencing datasets (90.6 Gb data). The superior computational performance of pseudoalignment-based methods over alignment-based methods, especially on large data sets, is important given the rapid increase of large genomic and metagenomic datasets in public databases. We hope this clarifies the impact of our approach compared to the existing state-of-the-art.

Furthermore, even though our analysis is focused on “gold-standard” complete genomes in NCBI RefSeq, the PCN dataset in our work is still ~1.8x and ~5x larger than the datasets reported by Ramiro-Martinez et al. and Shaw et al. respectively. We report PCN for ~11,338 plasmids, compared to 6,327 plasmids in Ramiro-Martinez et al. and 2,292 plasmids in Shaw et al. Furthermore, our dataset spans sequenced Bacteria and Archaea, while the PCN datasets reported by Ramiro-Martinez et al. and Shaw et al. only cover enterobacterial species.

Major comments:

- The scaling law that gets more attention here, between PCN and plasmid size, has been known for a long time. For decades it has remained qualitative. But it was also quantified recently. For example, Shaw, *Science Advances*, 2021 shows the same exact scaling law in Figure 3A. This scaling law is not new and the relevant past papers are not cited.

Thank you for bringing the study by Shaw et al. to our attention. To place our work in context, we have added the following text to the Introduction:

[...] A qualitative inverse correlation between plasmid size and copy number was described for 11 plasmids found in a *Bacillus thuringiensis* strain¹². Importantly, an analysis of 2,292 enterobacterial plasmids by Shaw et al. (2021) also revealed a quantitative inverse correlation between plasmid size and copy number, even though this pattern was not discussed or further examined in that work¹³. Despite these earlier contributions, comprehensive data on the distribution of PCN across Bacteria and Archaea still do not exist. Therefore, it is not known whether a quantitative relationship between plasmid size and copy number holds across Bacteria and Archaea. [...]

We agree with Reviewer 1 that Figure 3A in Shaw et al. shows a quantitative inverse correlation between plasmid size and copy number; indeed, our data shows that the relationship seen in these data generalizes across microbial taxa. In particular, Shaw et al. describe their result in Figure 3A as follows:

"Plasmids fell into two broad classes across genera: small multicopy plasmids (<10 kbp, 10× to 100× copy number inferred from coverage relative to chromosome) and large low-copy plasmids (>10 kbp, <10×) (Fig. 3A). AMR plasmids were almost all large low-copy plasmids (172 of 183, 94.0%). While small multicopy plasmids are of interest in facilitating evolutionary innovation (37, 38), this finding suggests that they do not play a major, direct role in AMR in livestock- and WwTW-associated Enterobacteriaceae."

Shaw et al. did not mention or describe a scaling law, correlation, or regression between plasmid size and copy number in their paper. Rather, they interpret their data in Figure 3A as showing an association between AMR and large low-copy number plasmids. Thus, while the data in Shaw et al. did indicate an inverse correlation between plasmid length and copy number (Figure 3A), the authors did not examine this relationship in depth or discuss its potential significance. In retrospect, their data, while analyzed in a smaller scale and less thoroughly, further underscores the robustness of the scaling law we describe in our study. In addition, the other two scaling laws in our paper (the protein-coding sequence scaling law and the metabolic gene scaling law) distinguish our study from those by Shaw et al. and Ramiro-Martinez et al.

- The fact that the number of genes correlates well with plasmid size is also well-known from works with over 20 years. Even the detail that for small size plasmids the gene density is lower was reviewed in reference #31 more than a decade ago (page 437).

Thank you for bringing this to our attention. We now put our findings in the context of this prior finding:

Protein-coding sequence scaling law. Smillie et al. (2010) reported that larger plasmids have protein-coding densities approaching that of chromosomes, while small plasmids are less coding dense. Our data shows that this pattern is universal across environments and microbial taxa (CITATION TO Smillie 2010 review "Mobility of Plasmids"). As plasmids increase in size, the fraction of sequence dedicated to protein-coding sequences converges to the fraction of sequence dedicated to protein-coding sequences on chromosomes (Figure 2A and 2B).

- The text of the manuscript has several problems. Some examples here.

We have revised our manuscript based on the specific comments raised below by Reviewer 1.

The results section has many details that should be in Methods.

We have moved the methodological details in the Results to the Methods.

The discussion needs improvement as it does not really discuss the biological reasons justifying the scaling laws and lacks references.

We have expanded the Discussion to include our working hypotheses for the biological causes for these scaling laws.

Many paragraphs are too small, down to less than two lines.

We have merged all small paragraphs into either the preceding or following paragraphs.

Section in page 10 pertains to discussion not results.

We have moved this section into the Discussion.

Section "A scalable bioinformatics pipeline for PCN estimation" is not really about Results.

We have revised the section “A scalable bioinformatics pipeline for PCN estimation” to focus on benchmarking results that demonstrate the computational superiority of the pseudoalignment approach for estimating plasmid copy number, compared to existing methods. The description of the pipeline in this section has been moved to Methods.

Novel results only really start at page 3 of the Results section.

Two aspects of our bioinformatics pipeline are novel contributions that belong in the Results. 1) The use of pseudoalignment for scalable PCN estimation. 2) The development of the PIRA algorithm to incorporate reads that map to multiple locations in the genome. We have revised the first two pages of the Results to focus on these two aspects, and we have moved all other routine details into the Methods. Pseudoalignment and PIRA have potential applications to biological questions beyond plasmid copy number estimation, as copy number changes are important in eukaryotic genome evolution and in cancer evolution. We now make this point in the Discussion.

The summary paragraph at the end of discussion seems redundant.

We have cut this final summary paragraph.

- The authors propose a method to identify PCN and compare it with other methods based on the same principle (sequence coverage). The difference between methods does not seem very large (Figure S1). There are some potential advantages of this method for lower PCN. But if the authors could provide coefficients of correlation for the panels in figure S1, they would reveal very high correlation. A related point is that there is no information on whether this method or the others are right. PCN was identified experimentally for some plasmids and it would be pertinent to know how they compare with published experimental data. By the end of page 17 the authors mention of a comparison with literature, but this table is never cited in results. When I went to the see the table I found that for most of these plasmids there is no information for PCN, which makes it not very useful. Also, for those plasmids for which there are numbers on PCN, the numbers are not given (nor the comparison with the computed PCN).

We have revised this section to clarify the issues raised here. Reviewer 1 is correct in noting that in most cases, the difference between methods is not large. Indeed, this finding is an important positive control that shows that our methods give sensible estimates in line with previous methods. This fact implies the robustness of the resulting scaling law to computational methodology. Further, even when PCN estimates are largely consistent across methods, having a more robust and rigorous analysis (through the use of PIRA) is beneficial because this generates a higher quality baseline quantification of PCNs compared to methods that neglect multireads. Finally, our benchmarks show that our method is algorithmically more efficient than existing methods, given the strong performance of pseuPIRA on large genomic datasets (Supplementary Table S2).

As suggested, we have included correlation coefficients in the S1 Figure legend to show this point. This finding shows that multireads usually have small effect on PCN estimation by short read mapping, when large numbers of reads (> 10,000 reads) map uniquely to the plasmids. In this regime, pseudoalignment provides PCN estimates that are as accurate as previous approaches, but much faster on large genomic datasets (Supplementary Table S2).

Multireads become more important when unique plasmid sequencing coverage is low. When the number of multireads is relatively large compared to the number of unireads, then failing to incorporate multireads (per existing methods that use sequencing coverage to estimate PCN) downwardly biases PCN estimates. In this regime, PIRA provides more sensible PCN estimates that incorporate all available information by including both multireads and unireads. We now make this point in Supplementary Figure S1 by plotting PCN for plasmids in which < 10,000 unireads are mapped using the naïve approach.

Reviewer 1 is also correct in noting that the analysis comparing sequencing-based approaches to PCN does not in itself demonstrate which set of estimates are more accurate. Nonetheless, our analysis of PCN for plasmids in which few unireads are mapped using the naïve approach (we use a threshold of 10,000 reads in Supplementary Figure S1A) shows that incorporating multireads provides more sensible PCN estimates for the set of plasmids that would otherwise have too few mapped reads. We have also included a comparison with PCN estimates for the plasmids reported by Shaw et al. (2021) in Supplementary Figure S1 and provide a simple performance comparison of the pseudoalignment approach against the current state-of-the-art (CoverM) in Supplementary Table S2. Note that the Table on page 17 is supporting evidence for our claim that plasmid copy numbers are rarely reported in the microbial genomics literature (i.e., it is standard for plasmid DNA sequences to be reported, but not for their copy number relative to chromosome to be reported in the literature).

In addition, we have revised our manuscript to clarify the following point, which was confusing in our submitted manuscript. Our analysis pipeline uses two novel improvements for estimate plasmid copy number: pseudoalignment for rapid PCN estimation, and the PIRA algorithm to calibrate those estimates by incorporating multiread information. As originally written, “PIRA” was confusingly used to refer to both the entire pipeline as well as the algorithm for incorporating multiread information. We now refer to our pipeline as “pseuPIRA” for Pseudoalignment and Probabilistic Iterative Read Assignment”.

- In page 8 there is a description of the fit of the model. But several important points seem missing. 1) if one has a plasmid of a given size, what is the expected interval of confidence for its copy number? This is a very relevant question that could be answered by this study. 2) How much of the variance is it explained by the scaling law? 3) What is the formula of the curve? This can be used to at least computed the expected PCN (although knowing the interval of confidence would be better).

We now report summary statistics for plasmids binned by percentiles by length in Supplementary Data File 3, including normally distributed 95% confidence intervals for plasmid copy numbers as length varies, also presented below.

Supplementary Figure S2. Summary statistics for plasmids binned by percentiles by length. Normally distributed 95% confidence intervals around the mean for each percentile by length are

in black. The mean PCN for each percentile by length are in red, and the Q25 and Q75 PCN quantiles for the percentiles by length are in blue.

In addition, we now report the amount of variance explained by the segmented regression along with the AIC reporting in the manuscript ($R^2 = 0.69$) and provide the formula for our segmented regression model in the notes for Supplementary Table S3, where we report the details of the segmented regression.

Supplementary Table 3. Details of segmented regression models.

Regression formula*	β_0	β_1	β_2	ψ
$\log_{10}(\text{copy number}) \sim \beta_0 + \beta_1(\log_{10}(\text{length})) + \beta_2(\log_{10}(\text{length}) - \psi) \times I(\log_{10}(\text{length}) > \psi)$	4.5606	-0.9315	0.7754	4.75
$\log_{10}(\text{copy number}) \sim \beta_0 + \beta_1(\log_{10}(\text{normalized length})) + \beta_2(\log_{10}(\text{normalized length}) - \psi) \times I(\log_{10}(\text{normalized length}) > \psi)$	-1.4744	-0.8905	0.7697	-1.761

* Here, $I(\cdot)$ is the indicator function equal to one when the statement is true. Here, β_1 is the left slope of the segmented regression, β_2 is the difference-in-slopes and ψ is the breakpoint. See Muggeo³⁹ for further details about this parameterization. The second model uses plasmid length normalized by the length of the largest chromosome in the genome.

Minor comments:

- "By contrast, mobilizable plasmids (i.e., those that can be transferred by conjugation, but do not themselves encode conjugation machinery) and non- mobilizable plasmids may be either large or small (Figure 1B)." This is well known.

We agree this property is likely well known to scientists working on MGEs. Our intent was to present it in a way that is accessible to a broad audience, who may not appreciate the extent of the wide distribution. We have revised the language to be more specific.

- Page 10 last paragraph. I could not understand why this explanation justifies why high copy number plasmids were more frequent in human-impacted environments.

We have rewritten this for clarity as follows.

Together, these observations suggest that small multicopy plasmids tend to have stable copy numbers of ~10-40 copies per chromosome per cell (Figure 1), such that plasmids with very high copy numbers (PCN > 50) may be signatures of recent positive selection for higher plasmid gene expression³.

- Page 11. "This finding implies that as plasmids increase in length, protein coding becomes more efficient." I don't understand this statement. What does it mean exactly that protein coding becomes more efficient?

We have edited this sentence to:

This finding implies that as plasmids decrease in length, protein coding density also decreases, as previously indicated by Smillie et al. (2010).

- Discussion. "Together, our results demonstrate the existence of fundamental constraints that dictate stable plasmid coding structures" Actually, the variance is quite high. The analysis of figure 1 reveals that PCN varies by two orders of magnitude for the same normalised length. I would suggest to tone down the mention to "fundamental constraints".

Thanks for raising this point. We now explicitly write in the Discussion that the analysis of Figure 1 reveals that PCN varies by two orders of magnitude for the same normalized length (for small plasmids), and that this variance in plasmid copy number decreases as plasmid length increases. In the sentence flagged by Reviewer 1, we have changed “fundamental” to “biophysical or evolutionary”. Our interpretation is that there must be some biophysical or evolutionary constraints (say bioenergetic or metabolic factors) that 1) constrain the copy number of large plasmids 2) select for lower coding density for small plasmids, and 3) select for increased numbers of metabolic genes, once plasmids pass some critical length threshold.

- In methods. Please provide the date of accession to the database.

We now provide the date of accession at which the data were last downloaded from NCBI RefSeq (March 28 2024).

- First paragraph in page 15 is hard to follow. I suggest a more detailed text and the inclusion of mathematical formulae.

We now provide more detailed description of how and why PIRA works, including mathematical formulae in a following section, “*Mathematical underpinnings of PIRA*”:

- In page 18. " The fraction of protein-coding sequences per replicon was calculated by summing up the length of each protein in each replicon (i.e., plasmid or chromosome) in each genome, and dividing by the length of that replicon" The length of each gene (not protein), I assume.

Yes, thanks for this catch. We have edited “length of each protein in each replicon” to “length of each protein-coding sequence in each replicon”.

- It would be nice if the program had a downloadable example to be run that did not imply downloading 15 Tb of data (in present conditions, I couldn't do it to make a test).

We have created a standalone command-line program `pseuPIRA.py` with an example that the reader can use to run our code.

Reviewer #1 (Remarks on code availability):

The presentation/documentation of the programs could be improved. The authors should also provide a toy example to test the program.

We have improved the presentation and documentation of our pipeline, we now include a standalone command-line program `pseuPIRA.py` with an example genome that the reader can use to test our program (<https://github.com/rohanmaddamsetti/pseuPIRA>)

Reviewer #2 (Remarks to the Author):

Scaling laws of plasmids across the microbial tree of life

Maddamsetti et al. have developed a sensible framework for plasmid copy number evaluation and applied it to a wide range of publicly available microbial sequences. Overall I very much enjoyed reading this paper and the complementary companion paper by Ramiro-Martínez et al.

Thank you for your warm comments.

My one major query reading Maddamsetti et al. is that the text implies that their PIRA pipeline is a bioinformatic tool that others can apply to their own data, when in fact it appears to be a hard-coded pipeline to reproduce the results of the manuscript (something I think few will attempt, given the large resource requirements the authors point out in their methods). This work would be much more impactful if the PIRA pipeline was available for others to apply to their own data. The github repo (<https://github.com/rohanmaddamsetti/PCN-db-pipeline>) should also have more descriptive documentation of the pipeline itself - a graphic of the workflow and tools used would be useful.

We have improved the presentation and documentation of our program, including workflow and tools used. In addition, we now provide a standalone command-line program, `pseuPIRA.py`, so that others can re-use and test our code more easily. We now describe the workflow and tools needed for PCN estimation here: <https://github.com/rohanmaddamsetti/pseuPIRA#Overview>. In addition, we divided the source code for the hardcoded PCN pipeline into two files: `PCN_library.py`, which contains all classes and functions, and `PCN_pipeline.py`, which just contains the stages of the pipeline. Each stage of the pipeline is described in the comments in the source code file `src/PCN_pipeline.py`; we now state this more clearly in the README here: <https://github.com/rohanmaddamsetti/PCN-db-pipeline#Overview>.

Minor comments:

Abstract (and intro):

"The capacity of a plasmid to express genes is constrained by two parameters: length and copy number."
- The phrasing of this sentence implies that gene expression is only due to length and copy number, however gene expression is also controlled by a number of other factors. I would rephrase to "plasmid length and copy number have a substantial impact on plasmid gene expression"

Thanks for the suggestion. We have made this change.

page 3, paragraph 1: "as they can engineered" -> typo, "as they can be engineered"

Thanks. We have fixed this typo.

Page 7, paragraph 3: Please provide range and SD for small and large plasmid cluster copy numbers and lengths.

We now provide the range and SD for small and large plasmid cluster copy numbers and lengths in the main text and in Supplementary Data File 4.

Page 7, paragraph 4: "Furthermore, this inverse correlation between PCN and length largely holds within individual genomes as well."

For the genomes that had positive correlations between length and copy number - was there any insight into why this was? This was between 10-20% of the sample size, so not infrequent. Was there anything interesting about the microbial species/source or plasmid type?

We found that most of these cases are genomes that only contain large plasmids. We now write up a deeper analysis in the manuscript as follows:

The intragenomic inverse correlation between PCN and length is even stronger when we consider the 2,008 genomes that contain at least one small plasmid. 1,649 of these genomes contain two or more plasmids, and 1,581 of those have an inverse correlation between plasmid length and copy number (mean Pearson correlation coefficient = -0.93), while 68 show a positive correlation (mean Pearson correlation coefficient = 0.76). 1,230 of these genomes contain three or more plasmids, and 1,199 of those have an inverse correlation between plasmid length and copy number (mean Pearson correlation coefficient = -0.90), while 31 show a positive correlation (mean Pearson correlation coefficient = 0.47). This analysis suggests that the genomes with positive intragenomic correlations between plasmid length and copy number are largely genomes that only contain large plasmids. It is possible that many such positive correlations could occur by chance, assuming that these plasmids largely have PCN ~ 1 .

page 9: "length is more conserved than copy number within PTUs" Kmer, ANI and mash distances are very dependent on length, so I find this conclusion somewhat circular. While this statement may be true based on this analysis, I am not sure it is reflective of any meaningful biological interpretation.

Yes, this is true. Our analysis taught us that Kmer, ANI, and mash distances were highly length-dependent (this is not well-reported in the plasmid literature), and we wanted to share this important methodological issue with a broader audience. See our response to the related point just below.

The last sentence, "This finding indicates that highly related plasmids have similar lengths, but that the copy numbers for small plasmids can vary over an order of magnitude"

Again, 'highly related' is based on the classification system being used, which here is mostly methods that rely on conserved sequence similarity and would favour similar length plasmids (and is again a bit of a circular argument). The second half of the sentence doesn't address any conclusion to do with PTUs.

This is a great point. We believe that this fact is largely underappreciated by microbiologists as we did not realize how length-dependent Kmer, ANI, and mash distance metrics were, until we conducted this analysis. We have rewritten the last sentence of this section to bring up the point raised by Reviewer 2:

This finding indicates that the copy numbers for small plasmids can vary over an order of magnitude, while highly related plasmids (as defined by K-mer similarity, and ANI) have similar lengths (Supplementary Figure S4). It is likely that the conservation of plasmid length within PTUs is a technical artifact caused by metrics that are highly sensitive to plasmid length (such as K-mer similarity and ANI) to define PTUs. Regardless, our analysis shows that the universal inverse power-law correlation between plasmid length and PCN holds across PTUs as defined by several different methods.

In addition, we have changed the title of this section to "*The inverse power-law between plasmid length and PCN holds across plasmid taxonomic units*" since this is a more biologically meaningful conclusion.

page 15: was there any QC of the sequencing reads for quality? Noting that PIRA does not take read quality into account.

We do not take sequencing read quality into account; one benefit of pseudoalignment is that with very high probability, reads with errors and sequencing artifacts such as adapters or barcodes will fail to pseudoalign to any replicon and are omitted from the analysis (Bray et al. 2016, Alanko et al. 2023). We now make this point in the Methods.

Page 15: the github repo (<https://github.com/rohanmaddamsetti/PCN-db-pipeline>) only lists Kallisto as a dependency, but Thermisto was used according to the manuscript? Please clarify. Please also list the tool dependencies and their version in the github.

Thanks for this catch. We have fixed this error in our documentation. We used Themisto for our PCN estimates and used kallisto for control experiments to show that PCN estimates are robust to the choice of pseudoalignment software. We now list the tool dependencies and their versions in the github documentation.

Page 18: Protein-coding sequence calculations: did this account for overlapping genes?

We have updated our calculations to explicitly check and account for overlapping genes (the effect is negligible).

Page 18: typo: "The, the union of all KEGG KO IDs"

Thanks, we have fixed this typo.

Sup table 1: URL should be changed to DOI or pubmed ID, URL may become inactive over time

We have updated these URLs to Pubmed IDs as suggested.

figure S6 (and other similar figures): add key on figure for green/orange/blue dots

We have added a legend for green/orange/blue dots to Figure S6 and similar figures as suggested.

Reviewer #2 (Remarks on code availability):

As above in main comments section. The code is available for reproducibility purposes, but not made to be easily accessible. The formatting of the readme in <https://github.com/rohanmaddamsetti/plasmid-scaling-laws> could be improved to help users follow the instructions. The instructions often refer to the authors host HPC, which is fine but should be distinguished from more general instructions to the wider community.

The PIRA pipeline (<https://github.com/rohanmaddamsetti/PCN-db-pipeline>) is not constructed as a stand-alone tool but as a hard-coded pipeline that will limit its accessibility to the community.

We have improved the presentation and documentation of our pipeline, and we now provide a standalone command-line program `pseuPIRA.py` (<https://github.com/rohanmaddamsetti/pseuPIRA>) so that others can re-use and test our code more easily.

Reviewer #3 (Remarks to the Author):

The authors need to take into account the following major points:

#) sequencing data

The authors stated, "Applying PIRA to all microbial genomes in the NCBI RefSeq database with linked short-read sequencing data in the Sequencing Read Archive (SRA), we analyzed 4,317 bacterial and archaeal genomes encompassing 11,338 plasmids, spanning the microbial tree of life."

However, if the authors are unsure about the methods used to obtain these sequencing data (e.g., bacterial growth conditions and DNA extraction methods used), the reliability of the results could be compromised. This issue should be addressed or, at the very least, discussed.

Below are relevant excerpts from a reference paper:

<https://www.sciencedirect.com/science/article/pii/S2001037018301685>

Plasmid size may be associated with copy number. For 11 plasmids found in *Bacillus thuringiensis* strain YBT-1520, the plasmid sizes (ranging from 2 to 416 kb) and the copy numbers determined by quantitative polymerase chain reaction (ranging from 1.38 to 172) were negatively correlated [94].

Plasmid copy number estimates can vary, depending on bacterial growth conditions and DNA extraction methods used [97,99]. Therefore, copy number data should be interpreted carefully.

Thank you for raising this critical point. We now discuss this issue in the Discussion, referencing the papers that Reviewer 3 has brought up, as follows:

Understanding why plasmid copy numbers shows such variability around the inverse scaling law remains an open question in need of further investigation. Significant variability in PCN estimates may be caused by differences in bacterial growth conditions as well as DNA extraction methods^{14,62}, and for this reason, we note that any particular PCN estimate in our dataset should be interpreted with care. Regardless, we expect our finding of an inverse scaling law between PCN to plasmid length be robust, given that it holds across 11,338 PCN estimates sampled across diverse bacteria and environments, and collected by diverse research groups using a range of experimental protocols for bacterial growth and DNA sequencing.

#) PIRA accurately estimates PCN

I suggest that the authors validate the accuracy of the PIRA-derived plasmid copy number (PCN) estimates by comparing them to the PCN values reported in previous studies, such as those cited below and in Supplementary Table 1:

Supplementary Table 1. Three out of fifty randomly chosen genomes containing plasmids have publications with reported plasmid copy numbers (PCN).

<https://academic.oup.com/bioinformatics/article/32/22/3380/2525610>

plasmidSPAdes: assembling plasmids from whole genome sequencing data

The average copy numbers (estimated as coverage ratios) varied from 1.4 for Bce to 14.0 for Cfr.

Thanks for this suggestion. We have included additional benchmarking in our paper against the independent PCN dataset in Shaw et al. (2021) in Science Advances (the paper mentioned by Reviewer 1); see Supplementary Figure 1F and associated main text.

#) A universal inverse power-law correlation between plasmid length and PCN

In statistical analyses with multiple genomes (e.g., correlation between plasmid length and PCN), data points should not be assumed to be independent, as closely related lineages share many traits from their common ancestor.

Reference:

Joseph Felsenstein (1985) *Phylogenies and the Comparative Method*
<https://www.jstor.org/stable/2461605>

While the reviewer brings up a valid point, phylogenetic independent contrasts may not be appropriate to control for non-independence, because plasmid evolution dynamics are largely determined by recombination and horizontal gene transfer, and therefore do not correlate well with genomic phylogenies defined by bacterial (core) genomes. The high rates of recombination within and between plasmid backbones mean that it may not be biologically meaningful to use plasmid phylogenies to control for non-independence between related plasmids, since phylogenies may represent a misleading model of vertical descent and modification between plasmids, when the true history is more like a network involving recombination between related segments and plasmid backbones. This is an ongoing issue in the field; for instance, see this paper (<https://www.microbiologyresearch.org/content/journal/mgen/10.1099/mgen.0.001290>)

Our PTU-level analyses in Supplementary Figure demonstrate that the inverse power-law correlation between plasmid length and PCN holds across plasmid taxonomic units; that is, when we collapse clusters of highly related plasmids into single PTUs, we still recover the same general trend. To make this point more clear, we have changed the title of the section “Length is more conserved than copy number within plasmid taxonomic groups (PTUs).” to “The inverse power-law between plasmid length and PCN holds across plasmid taxonomic units” since this is a more biologically meaningful conclusion.

We reiterate that Reviewer 3 brings up a valid point about phylogenetic non-independence; we now raise this issue in the Discussion:

We also expect the inverse scaling law to be robust to phylogenetic non-independence between genomes as plasmids, because the inverse scaling law holds across PTUs (Supplementary Figure S4). That said, an ongoing technical challenge in the field is the development of methods that can explicitly take phylogenetic non-independence between plasmids into account, given the plastic and dynamic nature of plasmid evolution which is dominated by structural variation, recombination, and horizontal gene transfer over point mutations and vertical inheritance^{63,64}.

Here are some minor points:

#) tree of life

The authors repeatedly use the term 'tree of life' throughout the manuscript. However, while taxonomic groups are presented in the paper, no phylogenetic trees—the 'tree of life' itself—are shown. If a phylogenetic tree is not provided or not considered, the term 'tree of life' should not be used.

We have changed “microbial tree of life” to “Bacteria and Archaea” throughout.

Point-by-point response to reviewers' comments

Reviewer #1 (Remarks to the Author):

The authors did substantial work to improve the manuscript and tackle my comments. Thanks for that. This is an interesting contribution that is now clearer, better contextualized and the discussion is much richer.

Thank you. We very much appreciate your constructive feedback on our manuscript.

A few comments:

The level of quantification displayed along the results section is very variable and at places it is vague. One should not have to dig in the substantial supplementary material to have an idea of the quantities involved.

Understood, thanks for highlighting this.

Some examples (lines are approximative, since the PDF does not number all lines):

- Around line 165. Comparable performance is vague, and slightly misleading. "On a small genomic dataset (1.58 Gb data), pseuPIRA has comparable performance to CoverM. " And "On a large genomic dataset (90.6 Gb data), however, pseuPIRA is substantially faster". This should not have qualifications that are arguable, it should have numbers. On a small dataset pseudoPIRA is 28% slower, on a large dataset (where speed is more relevant) it is twice as fast.

We have edited this text as suggested:

On a small genomic dataset (1.58 Gb data), **pseuPIRA is 1.29× slower than CoverM (16.6 seconds versus 12.9 seconds)**, while on a large genomic dataset (90.6 Gb data), pseuPIRA is **1.67× faster (705.0 seconds versus 1175.7 seconds)**, demonstrating the superior computational scaling properties of pseudoalignment over alignment for PCN estimation (Supplementary Table S2).

- Line 175 "generates PCN estimates that are consistent". Please quantify with a correlation coefficient of something of the sort

We have updated each reference to Supplementary Figure S1 in the main text with the relevant correlation coefficients reported in the Supplementary Figure S1 legend.

- Line 195. "estimates generated by pseuPIRA are consistent with PCN estimates generated by traditional read alignment algorithms" Same comment

Done.

- Line 199 "estimates were highly consistent" same comment

Done.

- Line 201 "Finally, we compared PCN estimates generated by pseuPIRA to the previously published PCN estimates in the Supplementary Table 2 of Shaw et al " The result of the comparison is not mentioned in the main text.

We now state in the main text that the Pearson correlation = 0.997.

Other comments:

- Line 307. I was confused with this new text. The authors state explicitly that PTUs conservation of size is likely a technical artifact, but show no evidence whatsoever of that. Previous works using more accurate methods (not ANI) have clearly demonstrated that plasmids within PTU have much more similar sizes than between PTUs. Which makes sense since plasmids with similarity are expected to have more similar sizes even if variance may be high. The authors own work goes in the same direction. If the authors want to draw some caution on the discussion about clustering of plasmids, that's fine. But here it's a clear statement in the Results section that lacks any kind of evidence. This may create great confusion in for newcomers in the field. As this is peripheral to the work, I suggest removing this unsubstantiated claim here and add a line that clustering of plasmids is hard and should be subject to caution in discussion.

We agree that further discussion of this point is peripheral to this work. We have rewritten this sentence to avoid overclaiming:

It is likely that the conservation of plasmid length within PTUs, to some degree, is related to the fact that PTUs are often defined using metrics that are highly sensitive to plasmid length (such as *k*-mer similarity, mash distance, or ANI).

- Discussion is still missing indications that comparisons here are being done between programs and not with actual experimental data. And that this and other approaches are assuming that coverage is an accurate measure of PCN.

To address the Reviewer's point, we now cite a reference that directly compares qPCR and sequencing coverage methods for PCN estimates in the laboratory (Ilhan et al. 2019), and note that differences in PCN estimates due to differences in quantitative method bears on this analysis:

Significant variability in PCN estimates may be caused by differences in bacterial growth conditions, DNA extraction methods^{14,63}, and experimental protocol (e.g., qPCR or whole genome sequencing⁶⁴) so for this reason, we note that any particular PCN estimate in our dataset should be interpreted with care.

- There is now more emphasis on plasmids with PCN <1. But there is a lot of variance here. I think this is missing a serious statistical analysis. What is the % of plasmids with PCN significantly lower than one? This number (not just those that are <1) should be the basis of discussion (which is very interesting).

We appreciate the comment. While our data show that a substantial number of plasmids have estimated PCN < 1 (2127/11338 = 19%, rounding PCN ≥ 0.95 to 1), determining whether these values are statistically "significantly" lower than 1 is challenging. This analysis would require prior knowledge of the expected PCN distribution, which is unavailable. Rather than making strong assumptions, we have chosen to present the descriptive patterns in the data (see

Supplementary Data File 3) and explore potential biological interpretations for PCN < 1 in the Discussion.

- Sup Fig 1 is missing in the PDF I got.

Our apologies. We noticed a bug where the PDF conversion during manuscript upload can fail to properly process some figures (such as Supplementary Figure 1). Although we updated the manuscript to fix this bug, it appears this error was re-introduced when we split the manuscript into separate Main Text and Supplementary Information files. We have carefully checked the Nature system PDF conversion on our end to make sure all Figures have been processed properly in our resubmission.

- Line 475. The authors data does show a more nuanced view. Maybe this should be explained in a couple of lines, in order to make it explicit?

We have added the following sentences to the manuscript to explain this result:

However, our data point to a more nuanced picture (Supplementary Table S3 and Supplementary Figure S11). Suppose the inverse scaling relation between plasmid length and copy number has slope = -1 on a log-log scale. Then plasmid DNA content (plasmid copy number \times plasmid length) should be constant (i.e., a flat line with slope = 0 on a log-log scale). However, Supplementary Figure S11 shows that the segmented regression on (normalized) plasmid DNA content has a first slope of 0.109 and a second slope of 0.879 (Adjusted $R^2 = 0.315$). Therefore, further theory and quantitative experiments are needed to understand the biophysical and evolutionary origins of these empirical scaling laws.

Reviewer #1 (Remarks on code availability):

I couldn't install the program simply because pip install uv didn't work in my computer and all the rest depends on it (MacOS X).

We now link to the installation instructions for uv in the pseuPIRA README (<https://docs.astral.sh/uv/getting-started/installation/>). This page provides multiple methods for users to install uv, in case one method fails.

Reviewer #2 (Remarks on code availability):

the authors have responded to previous comments and created a standalone tool for PCN estimates. While I have not installed/tested the tool, the github is well documented.

Thank you. We very much appreciate your constructive feedback on our manuscript.

Reviewer #3 (Remarks on code availability):

I recommend that the authors ensure the code runs correctly across different computing environments (Linux, macOS, etc.) and update the README documentation accordingly.

Thank you for your constructive feedback on our manuscript, and for reporting this bug.

We have updated our pipeline so that it can be run in TEST_MODE on a small number of genomes on a macOS computer. We have also updated the README to state that the pipeline can be run on MacOS in TEST_MODE using the command “python PCN_pipeline.py”.

We have updated the README to state that the full PCN pipeline **must** be run on a Linux HPC system with a SLURM job manager using the “sbatch” command for the following reasons:

1) several stages of the pipeline submit thousands of HPC jobs in parallel to speed up computation, and this is not possible on a laptop.

2) the SRA data download is substantial— ~15TB of sequencing reads – and so a full download is not possible. We debugged our pipeline on MacOS using a smaller test set, and then scaled up the pipeline on a Linux HPC cluster using slurm.

We now document these important details in the README.

Also, thanks for reporting the “uv install” bug in the pseuPIRA installation guide: this should have been “uv sync”, and we have updated the pseuPIRA README to address this.

Below are the commands I executed and their outputs when testing the installation and setup instructions:

...

```
git clone https://github.com/rohanmaddamsetti/pseuPIRA
cd pseuPIRA/
```

```
# https://github.com/rohanmaddamsetti/pseuPIRA?tab=readme-ov-file#requirements
```

```
$uname -m
arm64
```

```
wget https://github.com/algbio/themisto/releases/download/v3.2.2/themisto-v3.2.2-aarch64-apple-darwin22.tar.gz
tar xvzf themisto-v3.2.2-aarch64-apple-darwin22.tar.gz
cp themisto-v3.2.2-aarch64-apple-darwin22/themisto ~/bin
```

```
# https://github.com/rohanmaddamsetti/pseuPIRA?tab=readme-ov-file#dependency-management-with-uv
```

```
$pip install uv
Collecting uv
Downloading uv-0.7.2-py3-none-macosx_11_0_arm64.whl.metadata (11 kB)
Downloading uv-0.7.2-py3-none-macosx_11_0_arm64.whl (15.5 MB)
```

```
15.5/15.5 MB 18.3 MB/s
eta 0:00:00
```

```
Installing collected packages: uv
Successfully installed uv-0.7.2
```

```
$uv install
```

error: unrecognized subcommand 'install'

tip: a similar subcommand exists: 'uv pip install'

Usage: uv [OPTIONS] <COMMAND>

For more information, try '--help'.

...

...

git clone <https://github.com/rohanmaddamsetti/PCN-db-pipeline>

<https://github.com/rohanmaddamsetti/PCN-db-pipeline?tab=readme-ov-file#setup>

mkdir my-test-PCN-project

cd my-test-PCN-project

mkdir -p {data,results,src}

curl https://ftp.ncbi.nlm.nih.gov/genomes/GENOME_REPORTS/prokaryotes.txt > data/prokaryotes.txt ## on mac and linux

grep "plasmid" data/prokaryotes.txt | grep "Complete Genome" | sed 's/GCA/GCF/g' > results/complete-prokaryotes-with-plasmids.txt

conda create --name PCNdb_env --clone base

conda activate PCNdb_env

pip install pysradb biopython HTSeq beautifulsoup4 polars

conda install -c bioconda kallisto breseq

#Proceed (y/n)?

conda install -c conda-forge ncbi-datasets-cli --yes

Install SRA-Toolkit:

<https://github.com/rohanmaddamsetti/PCN-db-pipeline?tab=readme-ov-file#running-the-pipeline>

conda activate PCNdb_env

cd src/

sbatch --mem=16G -t 430:00:00 -p youlab --wrap="python PCN_pipeline.py"

-bash: sbatch: command not found

...

Thank you for this valuable reporting and feedback on running our code.

In this round of revision, two co-authors (independent from the main authors of the codebase) extensively tested our pipeline and pseuPIRA code on Linux HPC (Duke Compute Cluster and MIT Compute Cluster), while the lead author and a third co-author extensively tested both codes on MacOS and Linux HPC. The most common source of errors is dependency management, so please carefully read the README when setting up and running our code. Themisto and minimap2 must be installed and available on the command-line. In the github repositories, we now distribute a Linux binary for minimap2 and Linux and MacOS binaries for Themisto, to make it easier for users to install the required external software for pseuPIRA and the pipeline. In addition, SRA-Toolkit must be installed and configured properly to download fastq read data from NCBI servers. Finally, we found that polars version 1.30.0 introduces breaking changes to the polars API that causes PIRA code written with polars 1.24.0 to fail. Therefore, it is critical to install polars version 1.24.0 for pseuPIRA and the PCN-db-pipeline to work. These issues are

documented in both READMEs, and the uv package manager for pseuPIRA installation specifically installs compatible versions of python dependencies. Although these details are documented in the README, we are re-stating these points here, as these are common troubleshooting points that testers may encounter.

We greatly appreciate the reviewers' feedback, and we appreciate any additional reporting of any bugs or issues on the github repositories for pseuPIRA and the PCN-db-pipeline, so that others can build on our work.